# MITIGATING SUBOPTIMALITY OF DETERMINISTIC POLICY GRADIENTS IN COMPLEX Q-FUNCTIONS

## ABSTRACT

In reinforcement learning, off-policy actor-critic approaches like DDPG and TD3 are based on the deterministic policy gradient. Herein, the Q-function is trained from off-policy environment data and the actor (policy) is trained to maximize the Q-function via gradient ascent. We observe that in complex tasks like dexterous manipulation and restricted locomotion, the Q-value is a complex function of action, having several local optima or discontinuities. This poses a challenge for gradient ascent to traverse and makes the actor prone to get stuck at local optima. To address this, we introduce a new actor architecture that combines two simple insights: (i) use multiple actors and evaluate the Q-value maximizing action, and (ii) learn surrogates to the Q-function that are simpler to optimize with gradient-based methods. We evaluate tasks such as restricted locomotion, dexterous manipulation, and large discrete-action space recommender systems and show that our actor finds more optimal actions and outperforms alternate actor architectures.

## 1 INTRODUCTION

In sequential decision-making, the goal is to build an optimal agent that maximizes the expected cumulative returns (Sondik, 1971; Littman, 1996). Value-based reinforcement learning (RL) approaches estimate an action's future returns with a Q-value and select actions that maximize this Q-function (Sutton & Barto, 1998). However, in continuous action spaces, evaluating the Q-value of every possible action is impractical. This necessitates an actor, whose role is to globally maximize the Q-function, efficiently navigating the vast action space (Grondman et al., 2012). Yet, this is particularly challenging in tasks with a *non-convex* Q-function landscape with many local optima, such as restricted locomotion (Figure 2).

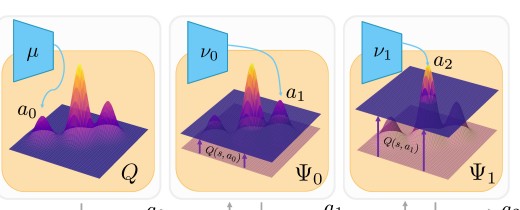

Figure 1: An actor $\mu$ trained with gradient ascent on a challenging Q-landscape gets stuck in local optima. Our approach learns a sequence of surrogates $\Psi_i$ of the Q-function that successively prune out the Q-landscape below the current best Q-values, resulting in fewer local optima. Thus, the actors $\nu_i$ trained to ascend on these surrogates produce actions with a more optimal Q-value.

Can we build an actor architecture to find better optimal actions in such complex Q-landscapes? Prior methods perform a search over the action space with evolutionary algorithms like CEM (De Boer et al., 2005; Kalashnikov et al., 2018; Shao et al., 2022), but this requires numerous costly re-evaluations of the Q-function. To avoid this, deterministic policy gradient (DPG) algorithms (Silver et al., 2014), such as DDPG (Lillicrap et al., 2015), TD3 (Fujimoto et al., 2018), and REDQ (Chen et al., 2020) train a parameterized actor to output actions with the objective of maximizing the Q-function locally.

A significant challenge arises in environments where the Q-function has many local optima, as shown in Figure 2. An actor trained via gradient ascent may converge to a local optimum with a much lower Q-value than the global maximum. This leads to *suboptimal* decisions during deployment and *sample-inefficient* training, as the agent fails to explore high-reward trajectories (Kakade, 2003).

To improve actors' ability to identify optimal actions in complex, non-convex Q-function landscapes, we propose the Successive Actors for Value Optimization (SAVO) algorithm. SAVO leverages two

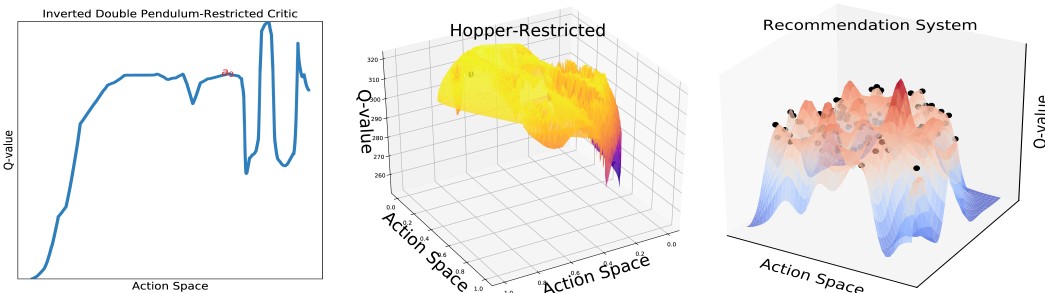

Figure 2: **Complex Q-landscapes.** We plot Q-value versus action $a$ for some state. In control of Inverted-Double-Pendulum-Restricted (left) and Hopper-Restricted (middle), certain action ranges are unsafe, resulting in various locally optimal actions. In a large discrete-action recommendation system (right), there are local peaks at actions representing real items (black dots).

key insights: (1) combining multiple policies using an $\arg\max$ on their Q-values to construct a superior policy (§4.1), and (2) simplifying the Q-landscape by excluding lower Q-value regions based on high-performing actions, inspired by tabu search (Glover, 1990), thereby reducing local optima and facilitating gradient-ascent (§4.2). By iteratively applying these strategies through a sequence of simplified Q-landscapes and corresponding actors, SAVO progressively finds more optimal actions.

We evaluate SAVO in complex Q-landscapes such as (i) *continuous* control in dexterous manipulation (Rajeswaran et al., 2017) and restricted locomotion (Todorov et al., 2012), and (ii) *discrete* decision-making in the large action spaces of simulated (Ie et al., 2019) and real-data recommender systems (Harper & Konstan, 2015), and gridworld mining expedition (Chevalier-Boisvert et al., 2018). We use the reframing of large discrete action RL to continuous action RL following (Van Hasselt & Wiering, 2009) and Dulac-Arnold et al. (2015), where a policy acts in continuous actions, such as the feature space of recommender items (Figure 2), and the nearest discrete action is executed.

Our key contribution is SAVO, an actor architecture to find better optimal actions in complex non-convex Q-landscapes (Section 4). In experiments, we visualize how SAVO's successively learned Q-landscapes have fewer local optima (Section 6.2), making it more likely to find better action optima with gradient ascent. This enables SAVO to outperform alternative actor architectures, such as sampling more action candidates (Dulac-Arnold et al., 2015) and learning an ensemble of actors (Osband et al., 2016) (Section 6.1) across continuous and discrete action RL.

## 2 RELATED WORK

Q-learning (Watkins & Dayan, 1992; Tesauro et al., 1995) is a fundamental value-based RL algorithm that iteratively updates Q-values to make optimal decisions. Deep Q-learning (Mnih et al., 2015) has been applied to tasks with manageable discrete action spaces, such as Atari (Mnih et al., 2013; Espeholt et al., 2018; Hessel et al., 2018), traffic control (Abdoos et al., 2011), and small-scale recommender systems (Chen et al., 2019). However, scaling Q-learning to continuous or large discrete action spaces requires specialized techniques to efficiently maximize the Q-function.

**Analytical Q-optimization.** Analytical optimization of certain Q-functions, such as wire fitting algorithm (Baird & Klopf, 1993) and normalized advantage functions (Gu et al., 2016; Wang et al., 2019), allows closed-form action maximization without an actor. Likewise, Amos et al. (2017) assume that the Q-function is convex in actions and use a convex solver for action selection. In contrast, the Q-functions considered in this paper are inherently non-convex in action space, making such an assumption invalid. Generally, analytical Q-functions lack the expressiveness of deep Q-networks (Hornik et al., 1989), making them unsuitable to model complex tasks like in Figure 2.

**Evolutionary Algorithms for Q-optimization.** Evolutionary algorithms like simulated annealing (Kirkpatrick et al., 1983), genetic algorithms (Srinivas & Patnaik, 1994), tabu search (Glover, 1990), and the cross-entropy method (CEM) (De Boer et al., 2005) are employed in RL for global optimization (Hu et al., 2007). Approaches such as QT-Opt (Kalashnikov et al., 2018; Lee et al., 2023; Kalashnikov et al., 2021) utilize CEM for action search, while hybrid actor-critic methods like CEM-RL (Pourchot & Sigaud, 2018), GRAC (Shao et al., 2022), and Cross-Entropy Guided Policies (Simmons-Edler et al., 2019) combine evolutionary techniques with gradient descent. Despite their effectiveness, CEM-based methods require numerous Q-function evaluations and struggle with

high-dimensional actions (Yan et al., 2019). In contrast, SAVO achieves superior performance with only a few (e.g., three) Q-evaluations, as demonstrated in experiments (Section 6).

**Actor-Critic Methods with Gradient Ascent.** Actor-critic methods can be on-policy (Williams, 1992; Schulman et al., 2015; 2017) primarily guided by the policy gradient of expected returns, or off-policy (Silver et al., 2014; Lillicrap et al., 2015; Fujimoto et al., 2018; Chen et al., 2020) primarily guided by the bellman error on the critic. Deterministic Policy Gradient (DPG) (Silver et al., 2014) and its extensions like DDPG Lillicrap et al. (2015), TD3 (Fujimoto et al., 2018) and REDQ (Chen et al., 2020) optimize actors by following the critic's gradient. Soft Actor-Critic (SAC) (Haarnoja et al., 2018) extends DPG to stochastic actors. However, these methods can get trapped in local optima within the Q-function landscape. SAVO addresses this limitation by enhancing gradient-based actor training. This issue also affects stochastic actors, where a local optimum means an *action distribution* (instead of a single action) that fails to minimize the KL divergence from the Q-function density fully, and is a potential area for future research.

**Sampling-Augmented Actor-Critic.** Sampling multiple actions and evaluating their Q-values is a common strategy to find optimal actions. Greedy actor-critic (Neumann et al., 2018) samples high-entropy actions and trains the actor towards the best Q-valued action, yet remains susceptible to local optima. In large discrete action spaces, methods like Wolpertinger (Dulac-Arnold et al., 2015) use k-nearest neighbors to propose actions, requiring extensive Q-evaluations on up to 10% of total actions. In contrast, SAVO efficiently generates high-quality action proposals through successive actor improvements without being confined to local neighborhoods.

**Ensemble-Augmented Actor-Critic.** Ensembles of policies enhance exploration by providing diverse action proposals through varied initializations (Osband et al., 2016; Chen & Peng, 2019; Song et al., 2023; Zheng12 et al., 2018; Huang et al., 2017). The best action is selected based on Q-value evaluations. Unlike ensemble methods, SAVO systematically eliminates local optima, offering a more reliable optimization process for complex tasks (Section 6).

## 3 PROBLEM FORMULATION

Our work tackles the effective optimization of the Q-value landscape in off-policy actor-critic methods for continuous and large-discrete action RL. We model a task as a Markov Decision Process (MDP), defined by a tuple $\{\mathcal{S}, \mathcal{A}, \mathcal{T}, R, \gamma\}$ of states, actions, transition probabilities, reward function, and a discount factor. The action space $\mathcal{A}$ is a $D$-dimensional *continuous* vector space, $\mathbb{R}^D$. At every step $t$ in the episode, the agent receives a state observation $s_t \in \mathcal{S}$ from the environment and acts with $a_t \in \mathcal{A}$. Then, it receives the new state after transition $s_{t+1}$ and a reward $r_t$. The objective of the agent is to learn a policy $\pi(a \mid s)$ that maximizes the expected discounted reward, $\max_\pi \mathbb{E}_\pi \left[ \sum_t \gamma^t r_t \right]$.

### 3.1 DETERMINISTIC POLICY GRADIENTS (DPG)

DPG (Silver et al., 2014) is an off-policy actor-critic algorithm that trains a deterministic actor $\mu_\phi$ to maximize the Q-function. This happens via two steps of generalized policy iteration, GPI (Sutton & Barto, 1998): policy evaluation estimates the Q-function (Bellman, 1966) and policy improvement greedily maximizes the Q-function. To approximate the $\arg\max$ over continuous actions in Eq. 2, DPG proposes the policy gradient to update the actor locally in the direction of increasing Q-value,

$$Q^\mu(s, a) = r(s, a) + \gamma \mathbb{E}_{s'} \left[ Q^\mu(s', \mu(s')) \right], \tag{1}$$

$$\mu(s) = \arg\max_a Q^\mu(s, a), \tag{2}$$

$$\nabla_\phi J(\phi) = \mathbb{E}_{s \sim \rho^\mu} \left[ \nabla_a Q^\mu(s, a) \big|_{a=\mu(s)} \nabla_\phi \mu_\phi(s) \right]. \tag{3}$$

DDPG (Lillicrap et al., 2015) and TD3 (Fujimoto et al., 2018) made DPG compatible with deep networks via techniques like experience replay and target networks to address non-stationarity of online RL, twin critics to mitigate overestimation bias, target policy smoothing to prevent exploitation of errors in the Q-function, and delayed policy updates so critic is reliable to provide actor gradients.

### 3.2 THE CHALLENGE OF AN ACTOR MAXIMIZING A COMPLEX Q-LANDSCAPE

DPG-based algorithms train the actor following the chain rule in Eq. 3. Specifically, its first term, $\nabla_a Q^\mu(s, a)$ involves gradient ascent in Q-versus-$a$ landscape. This Q-landscape is often highly non-convex (Fig. 2, 3) and changes non-stationarily during training. This makes the actor's output

$\mu(s)$ get stuck at suboptimal Q-values, thus leading to insufficient policy improvement in Eq. 2. We can define the suboptimality of the $\mu$ w.r.t. $Q^\mu$ at state $s$ as

$$\Delta(Q^\mu, \mu, s) = \arg \max_a Q^\mu(s, a) - Q^\mu(s, \mu(s)) \geq 0. \qquad (4)$$

Suboptimality in actors is a crucial problem because it leads to (i) **poor sample efficiency** by slowing down GPI, and (ii) **poor inference performance** even with an optimal Q-function, $Q^*$ as seen in Fig. 3 where a TD3 actor gets stuck at a locally optimum action $a_0$ in the final Q-function.

This challenge fundamentally differs from the well-studied field of non-convex optimization, where non-convexity arises in the *loss function w.r.t. the model parameters* (Good-fellow, 2016). In those cases, stochastic gradient-based optimization methods like SGD and Adam (Kingma & Ba, 2014) are effective at finding acceptable local minima due to the smoothness and high dimensionality of the parameter space, which often allows for escape from poor local optima (Choromanska et al., 2015). Moreover, overparameterization in deep networks can lead to loss landscapes with numerous good minima (Neyshabur et al., 2017).

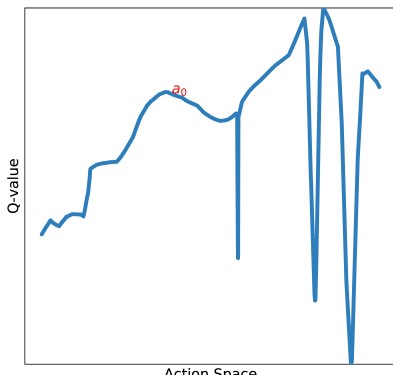

Figure 3: Non-convex Q-landscape in Inverted-Pendulum-Restricted leads to a suboptimally converged actor.

In contrast, our challenge involves non-convexity in the *Q-function w.r.t. the action space*. The actor's task is to find, for every state $s$, the action $a$ that maximizes $Q^\mu(s, a)$. Since the Q-function can be highly non-convex and multimodal in $a$, the gradient ascent step $\nabla_a Q^\mu(s, a)$ used in Eq. 3 may lead the actor to converge to suboptimal local maxima in action space. Unlike parameter space optimization, the actor cannot rely on high dimensionality or overparameterization to smooth out the optimization landscape in action space because the Q-landscape is determined by the task's reward. Furthermore, the non-stationarity of the Q-function during training compounds this challenge. These properties make our non-convex challenge unique, requiring a specialized actor to navigate the complex Q-landscape.

Tasks with several local optima in the Q-function include restricted inverted pendulum shown in Fig. 3, where certain regions of the action space are invalid or unsafe, leading to a rugged Q-landscape (Florence et al., 2022). Dexterous manipulation tasks exhibit discontinuous behaviors like inserting a precise peg in place with a small region of high-valued actions (Rajeswaran et al., 2017) and surgical robotics have a high variance in Q-values of nearby motions (Barnoy et al., 2021).

### 3.2.1 LARGE DISCRETE ACTION RL REFRAMED AS CONTINUOUS ACTION RL

We discuss another practical domain where non-convex Q-functions are present. In large discrete action tasks like recommender systems (Zhao et al., 2018; Zou et al., 2019; Wu et al., 2017), a common approach (Van Hasselt & Wiering, 2009; Dulac-Arnold et al., 2015) is to use continuous representations of actions as a medium of decision-making. Given a set of actions, $\mathcal{I} = \{\mathscr{I}_1, \ldots, \mathscr{I}_N\}$, a predefined module $\mathcal{R} : \mathcal{I} \to \mathcal{A}$ assigns each $\mathscr{I} \in \mathcal{I}$ to its representation $\mathcal{R}(\mathscr{I})$, e.g., text embedding of a given movie (Zhou et al., 2010). A continuous action policy $\pi(a \mid s)$ is learned in the action representation space, with each $a \in \mathcal{A}$ converted to a discrete action $\mathscr{I} \in \mathcal{I}$ via nearest neighbor,

$$f_{\text{NN}}(a) = \arg \min_{\mathscr{I}_i \in \mathcal{I}} \|\mathcal{R}(\mathscr{I}_i) - a\|_2.$$

Importantly, the nearest neighbor operation creates a challenging piece-wise continuous Q-function with suboptima at various discrete points as shown in Fig. 2 (Jain et al., 2021; 2020).

## 4 APPROACH: SUCCESSIVE ACTORS FOR VALUE OPTIMIZATION (SAVO)

Our objective is to design an actor architecture that efficiently discovers better actions in complex, non-convex Q-function landscapes. We focus on gradient-based actors and introduce two key ideas:

1. **Maximizing Over Multiple Policies:** By combining policies using an $\arg \max$ over their Q-values, we can construct a policy that performs at least as well as any individual policy (§4.1).

2. **Simplifying the Q-Landscape:** Drawing inspiration from tabu search (Glover, 1990), we propose using actions with good Q-values to eliminate or "tabu" the Q-function regions with lower Q-values, thereby reducing local optima and facilitating gradient-based optimization (§4.2).

Figure 4: **SAVO Architecture.** (left) Q-network is unchanged. (center) Instead of a single actor, we learn a sequence of actors and surrogate networks connected via action predictions. (right) Conditioning on previous actions is done with the help of a deep-set summarizer and FiLM modulation.

### 4.1 MAXIMIZER ACTOR OVER ACTION PROPOSALS

We first show how additional actors can improve DPG's policy improvement step. Given a policy $\mu$ being trained with DPG over $Q$, consider $k$ additional arbitrary policies $\nu_1, \ldots, \nu_k$, where $\nu_i : \mathcal{S} \to \mathcal{A}$ and let $\nu_0 = \mu$. We define a maximizer actor $\mu_M$ for $a_i = \nu_i(s)$ for $i = 0, 1, \ldots, k$,

$$\mu_M(s) := \underset{a \in \{a_0, a_1, \ldots, a_k\}}{\arg\max} Q(s, a), \qquad (5)$$

Here, $\mu_M$ is shown to be a better maximizer of $Q(s, a)$ in Eq. 2 than $\mu \ \forall s$ :

$$Q(s, \mu_M(s)) = \max_{a_i} Q(s, a_i) \geq Q(s, a_0) = Q(s, \mu(s)).$$

Therefore, by policy improvement theorem (Sutton & Barto, 1998), $V^{\mu_M}(s) \geq V^{\mu}(s)$, proving that $\mu_M$ is better than a single $\mu$ for a given $Q$. Appendix B proves the following theorem by showing that policy evaluation and improvement with $\mu_M$ converge.

**Theorem 4.1** (Convergence of Policy Iteration with Maximizer Actor). *A modified policy iteration algorithm where $\nu_0 = \mu$ is the current policy learned with DPG and maximizer actor $\mu_M$ defined in Eq. 5, converges in the tabular setting to the optimal policy.*

This algorithm is valid for arbitrary $\nu_1, \ldots \nu_k$. We experiment with $\nu$'s obtained by **sampling** from a Gaussian centered at $\mu$ or **ensembling** on $\mu$ to get diverse actions. However, in high-dimensionality, *randomness* around $\mu$ is not sufficient to get action proposals to significantly improve $\mu$.

### 4.2 SUCCESSIVE SURROGATES TO REDUCE LOCAL OPTIMA

To train additional policies $\nu_i$ that can improve upon $\mu_M$, we introduce *surrogate* Q-functions with fewer local optima, inspired by the principles of Tabu Search (Glover & Laguna, 1998), which is an optimization technique that uses memory structures to avoid revisiting previously explored inferior solutions, thereby enhancing the search for optimal solutions. Similarly, our surrogate functions act as memory mechanisms that "tabu" certain regions of the Q-function landscape deemed suboptimal based on previously identified good actions. Given a known action $a^\dagger$, we define a surrogate function that elevates the Q-values of all inferior actions to $q(s, a^\dagger)$, which serves as a constant threshold:

$$\Psi(s, a; a^\dagger) = \max\{Q(s, a), q(s, a^\dagger)\}. \qquad (6)$$

Extending this idea, we define a sequence of surrogate functions using the actions from previous policies. Let $a_{<i} = \{a_0, a_1, \ldots, a_{i-1}\}$. The $i$-th surrogate function is:

$$\Psi_i(s, a; a_{<i}) = \max\left\{Q(s, a), \max_{j<i} q(s, a_j)\right\}. \qquad (7)$$

**Theorem 4.2.** *For a state $s \in \mathcal{S}$ and surrogates $\Psi_i$ defined as above, the number of local optima decreases with each successive surrogate:*

$$N_{opt}(Q(s, \cdot)) \geq N_{opt}(\Psi_1(s, \cdot; a_0)) \geq \cdots \geq N_{opt}(\Psi_k(s, \cdot; a_{<k})),$$

*where $N_{opt}(f)$ denotes the number of local optima of function $f$ over $\mathcal{A}$.*

**Proof Sketch.** As $\Psi_i \to \Psi_{i+1}$, the anchor Q-value in Eq. 7 weakly increases, $\max_{j<i} Q(s, a_j) \leq \max_{j<(i+1)} Q(s, a_j)$, thus, eliminating more local minima below it (proof in Appendix C.1). □

### 4.3 SUCCESSIVE ACTORS FOR SURROGATE OPTIMIZATION

To effectively reduce local optima using the surrogates $\Psi_1, \ldots, \Psi_k$, we design the policies $\nu_i$ to optimize their respective surrogates $\Psi_i(s, a; a_{<i})$. Each $\nu_i$ focuses on regions where $Q(s, a) \geq \max_{j<i} q(s, a_j)$, allowing it to find better optima than previous policies. The actor $\nu_i$ is conditioned on previous actions $\{a_0, \ldots, a_{i-1}\}$, summarized using deep sets (Zaheer et al., 2017) (Figure 4). The maximizer actor $\mu_M$ (Eq. 5) selects the best action among all proposals.

We train each actor $\nu_i$ using gradient ascent on its surrogate $\Psi_i$, similar to DPG:

$$\nabla_{\phi_i} J(\phi_i) = \mathbb{E}_{s \sim \rho^{\mu_M}} \left[ \nabla_a \Psi_i(s, a; a_{<i}) \big|_a \nabla_{\phi_i} \nu_i(s; a_{<i}) \right]. \tag{8}$$

### 4.4 APPROXIMATE SURROGATE FUNCTIONS

The surrogates $\Psi_i$ have zero gradients when $Q(s, a) < \tau$, where $\tau = \max_{j<i} Q(s, a_j)$,

$$\nabla_a \Psi_i(s, a; a_{<i}) = \begin{cases} \nabla_a Q^{\mu_M}(s, a) & \text{if } Q(s, a) \geq \tau, \\ 0 & \text{if } Q(s, a) < \tau. \end{cases}$$

This means the policy gradient only updates $\nu_i$ when $Q(s, a) \geq \tau$, which may slow down learning. To address this issue, we ease the gradient flow by learning a smooth lossy approximation $\hat{\Psi}_i$ of $\Psi_i$.

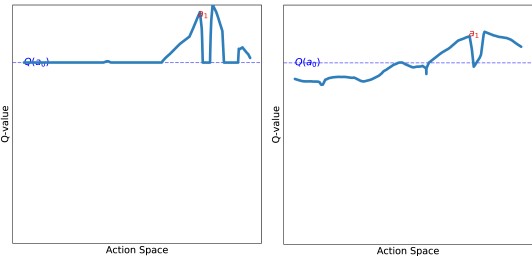

Figure 5: While $\Psi$ (left) has flat surfaces, $\hat{\Psi}$ (right) smoothens the function to allow non-zero gradients to flow into the actor towards better optima in Inverted-Pendulum-Restricted.

We approximate each surrogate $\Psi_i$ with a neural network $\hat{\Psi}_i$. This approach leverages the universal approximation theorem (Hornik et al., 1989; Cybenko, 1989) and benefits from empirical evidence that deep networks can effectively learn non-smooth functions (Imaizumi & Fukumizu, 2019). The smooth surrogate $\hat{\Psi}_i$ enables continuous gradient propagation, which is essential for optimizing the actors $\nu_i$. We train $\hat{\Psi}_i$ to approach $\Psi_i$ by minimizing the mean squared error at two critical points:

$$\mathcal{L}_{\text{approx}} = \mathbb{E}_{s \sim \rho^{\mu_M}} \left[ \sum_{a \in \{\tilde{\mu}_M(s), \nu_i(s; a_{<i})\}} \left\| \hat{\Psi}_i(s, a; a_{<i}) - \Psi_i(s, a; a_{<i}) \right\|_2^2 \right], \text{ where} \tag{9}$$

1. $\tilde{\mu}_M(s)$ represents a high Q-value action selected by the current maximizer actor $\mu_M$.
2. $\nu_i(s; a_{<i})$ is the action proposed by the $i$-th actor conditioned on previous actions $a_{<i}$.

This design ensures $\hat{\Psi}_i$ is updated on high Q-value actions and thus the landscape is biased towards those values. This makes the gradient flow trend in the direction of high Q-values. So, even when $a_i$ from $\nu_i$ falls in a region of zero gradients for $\Psi_i$, in $\hat{\Psi}_i$ would provide policy gradient in a higher Q-value direction, if it exists. Figure 5 shows $\Psi_1$ and $\hat{\Psi}_1$ in restricted inverted pendulum task. Figure 23 analyzes $\mathcal{L}_{\text{approx}}$ over training, demonstrating that $\hat{\Psi}_i$ stays close to $\Psi_i$ while smoothing it.

### 4.5 SAVO-TD3 ALGORITHM AND DESIGN CHOICES

While the SAVO architecture (Figure 4) can be integrated with any off-policy actor-critic algorithm, we choose to implement it with TD3 (Fujimoto et al., 2018) due to its compatibility with continuous and large-discrete action RL (Dulac-Arnold et al., 2015). Using the SAVO actor in TD3 enhances its ability to find better actions in complex Q-function landscapes. Algorithm 1 depicts SAVO (highlighted) applied to TD3. We discuss design choices in SAVO and validate them in Section 6.

1. **Removing policy smoothing**: We eliminate TD3's policy smoothing, which adds noise to the target action $\tilde{a}$ during critic updates. In non-convex landscapes, nearby actions may have significantly different Q-values and noise addition might obscure important variations.

2. **Exploration in Additional Actors**:
Successive actors $\nu_i$ explore their surrogate landscapes by adding OU (Lillicrap et al., 2015) or Gaussian (Fujimoto et al., 2018) noise to their outputs, effectively discovering high-reward regions.

3. **Twin Critics for Surrogates**:
To prevent overestimation bias in surrogates $\hat{\Psi}_i$, we use twin critics to compute the target of each surrogate, mirroring TD3.

4. **Conditioning on Previous Actions**:
Actors $\nu_i$ and surrogates $\hat{\Psi}_i$ are conditioned on preceding actions via FiLM layers (Perez et al., 2018) as in Fig. 4.

5. **Discrete Action Space Tasks**:
We apply 1-nearest-neighbor $f_{\text{NN}}$ before the Q-function, so it is only queried at in-distribution actions. For gradient flow into the actor, a noisy Q-function is added. See Q-smoothing in Section G.3.

---

**Algorithm 1** SAVO-TD3

Initialize $Q, Q_2, \mu, \quad \hat{\Psi}_1, \ldots, \hat{\Psi}_k, \quad \nu_1, \ldots, \nu_k$
Initialize target networks $Q' \leftarrow Q, Q_2' \leftarrow Q_{twin}$
Initialize replace buffer $\mathcal{B}$.
**for** timestep $t = 1$ to $T$ **do**
  **Select Action:**
  Evaluate $a_0 = \mu(s), a_i = \nu_i(s; a_{<i})$
  Add perturbations with OU Noise $\hat{a}_i = a_i + \epsilon_i$
  Evaluate $\mu_M(s) = \arg\max_{a \in \{\hat{a}_0, \ldots, \hat{a}_k\}} Q^\mu(s, a)$
  Exploration action $a = \tilde{\mu}_M(s) = \mu_M(s) + \epsilon$
  Observe reward $r$ and new state $s'$
  Store $(s, a, \{\hat{a}_i\}_{i=0}^K, r, s')$ in $\mathcal{B}$
  **Update:**
  Sample N transitions $(s, a, \{\hat{a}_i\}_{i=0}^K, r, s')$ from $\mathcal{B}$
  Compute target action $\tilde{a} = \mu_M(s')$
  Update $Q, Q_2 \leftarrow r + \gamma \min\{Q'(s', \tilde{a}), Q_2'(s', \tilde{a})\}$

  Update $\hat{\Psi}_i$ with Eq. 9 $\forall i = 1, \ldots k$
  Update actor $\mu$ with Eq. 3
  Update actor $\nu_i$ with Eq. 8 $\forall i = 1, \ldots k$
**end for**

---

SAVO-TD3 systematically reduces local optima through successive surrogates while leveraging TD3 as a robust RL baseline. In the next section, we validate these design choices through experiments, demonstrating SAVO-TD3's effectiveness in complex reinforcement learning tasks against alternate actor architectures.

# 5 ENVIRONMENTS

We evaluate SAVO on discrete and continuous action space environments with challenging Q-value landscapes. More environment details are presented in Appendix D and Figure 12.

**Locomotion in Mujoco-v4.** We evaluate Mujoco (Todorov et al., 2012) environments of Hopper, Walker2D, Inverted Pendulum, and Inverted Double Pendulum.

**Locomotion in Restricted Mujoco.** We create a restricted locomotion suite of the same environments as in Mujoco-v4. A *hard* Q-landscape is realized via high-dimensional discontinuities that restrict the action space. Concretely, a set of predefined hyper-spheres (as shown in Figure 6) in the action space are sampled and set to be valid actions, while the other invalid actions have a null effect if selected. The complete details can be found in Appendix D.3.1.

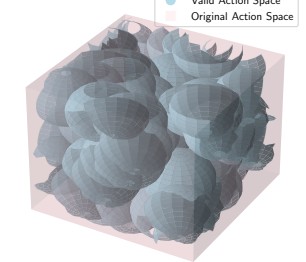

Figure 6: Hopper's 3D visualization of Action Space.

**Adroit Dexterous Manipulation** (Rajeswaran et al., 2017) *Door*: In this task, a robotic hand is required to open a door with a latch. The challenge lies in the precise manipulation needed to unlatch and swing open the door using the fingers. *Hammer*: the robotic hand must use a hammer to drive a nail into a board. This task tests the hand's ability to grasp the hammer correctly and apply force accurately to achieve the goal. *Pen*: This task involves the robotic hand manipulating a pen to reach a specific goal position and rotation. The objective is to control the pen's orientation and position using fingers, which demands fine motor skills and coordination.

**Mining Expedition in Grid World.** We develop a 2D Mining grid world environment (Chevalier-Boisvert et al., 2018) where the agent (Appendix Fig. 12) navigates a 2D maze to reach the goal, removing mines with correct pick-axe tools to reach the goal in the shortest path. The action space includes navigation and tool-choice actions, with a procedurally-defined action representation space. The Q-landscape is non-convex because of the diverse effects of nearby action representations.

**Simulated and Real-Data Recommender Systems.** RecSim (Ie et al., 2019) simulates sequential user interactions in a recommender system with a large discrete action space. The agent must recommend the most relevant item from a set of 10,000 items based on user preference information.

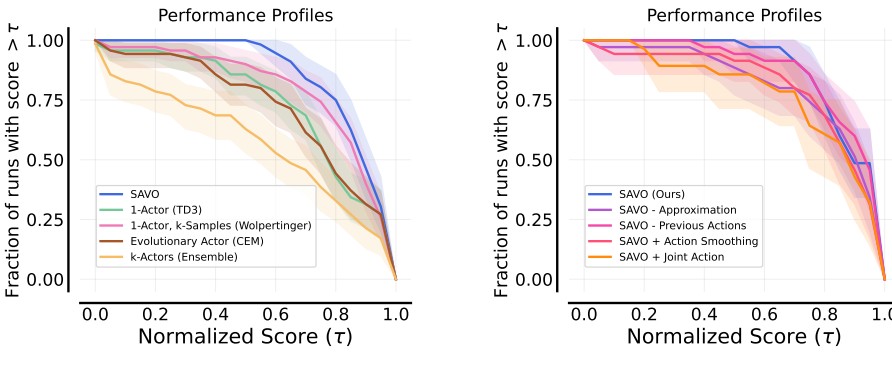

(a) SAVO versus baseline actor architectures.          (b) SAVO versus ablations of SAVO

Figure 7: Aggregate performance profiles using normalized scores over 7 tasks and 10 seeds each.

The action representations are simulated item characteristic vectors in simulated and movie review embeddings in the real-data task based on MovieLens (Harper & Konstan, 2015) for items.

## 6 EXPERIMENTS

### 6.1 EFFECTIVENESS OF SAVO IN CHALLENGING Q-LANDSCAPES

We compare SAVO against the following baseline actor architectures:

- **1-Actor (TD3)**: Conventional single actor architecture which is susceptible to local optima.
- **1-Actor, $k$=3 samples (Wolpertinger)**: Gaussian sampling centered on actor's output. For discrete actions, we select 3-NN discrete actions around the continuous action (Dulac-Arnold et al., 2015).
- $k$**=3-Actors (Ensemble)**: Each actor (Osband et al., 2016) can find different local optima, improving the best action.
- **Evolutionary actor (CEM)**: Repeated rounds of search with CEM over the action space (Kalashnikov et al., 2018).
- **Greedy-AC**: Greedy Actor Critic (Neumann et al., 2018) trains a high-entropy proposal policy and primary actor trained from best proposals with gradient updates.
- **Greedy TD3**: Our version of Greedy-AC with TD3 exploration and update improvements.
- **SAVO**: Our method with 3 successive actors and surrogate Q-landscapes.

We ablate the crucial components and design decisions in SAVO:

- **SAVO - Approximation**: removes the approximate surrogates (Sec. 4.4), using $\Psi_i$ instead of $\hat{\Psi}_i$.
- **SAVO - Previous Actions**: removes conditioning on $a_{<i}$ in SAVO's actors and surrogates.
- **SAVO + Action Smoothing**: TD3's policy smoothing (Fujimoto et al., 2018) adds action noise to compute Q-targets.
- **SAVO + Joint Action**: trains an actor with a joint action space of $3 \times D$. The $k$ action samples are obtained by splitting the joint action into $D$ dimensions. Validates successive nature of SAVO.

**Aggregate performance.** We utilize performance profiles (Agarwal et al., 2021) to aggregate results across different environments in Figure 7a (evaluation mechanism detailed in Appendix H.1). SAVO consistently outperforms baseline actor architectures like single-actor (TD3) and sampling-augmented actor (Wolpertinger), showing the best robustness across challenging Q-landscapes. In Figure 7b, SAVO outperforms its ablations, validating each proposed component and design decision.

**Per-environment results.** In Mining Expedition, the action space has semantically different navigation and tool-choice actions, while RecSim and RecSim-Data have a large and diverse set of items. The Q-landscape is significantly non-convex in such discrete tasks because the continuous action goes through a nearest-neighbor step to select a discrete item. Thus, sampling more neighbors in a local neighborhood via Wolpertinger is better than TD3's single action in Figure 8. However, the optimal action is not necessarily near the initial guess. Therefore, SAVO achieves the best performance by directly addressing global non-convexity. In restricted locomotion with a discontinuous action space, SAVO's actors can search far separated regions to optimize the Q-landscape better than only nearby

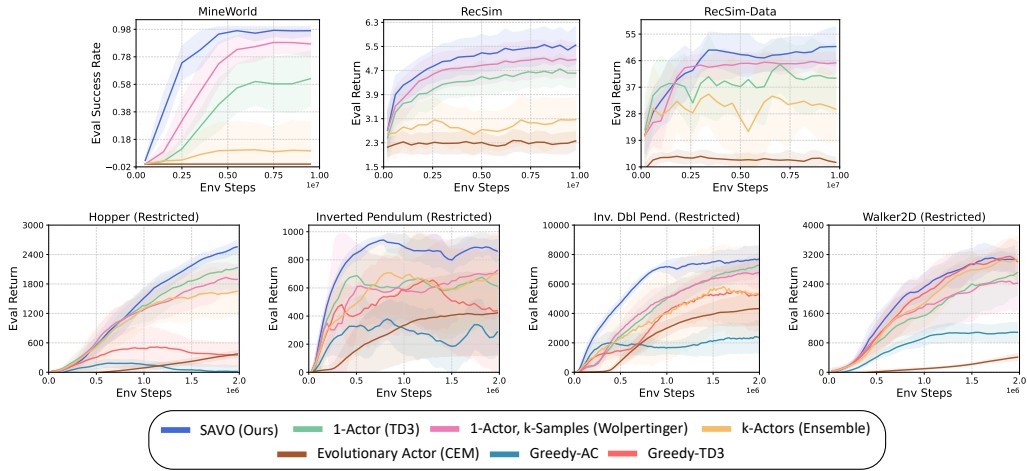

Figure 8: SAVO against baselines on discrete and continuous tasks. Results averaged over 10 seeds.

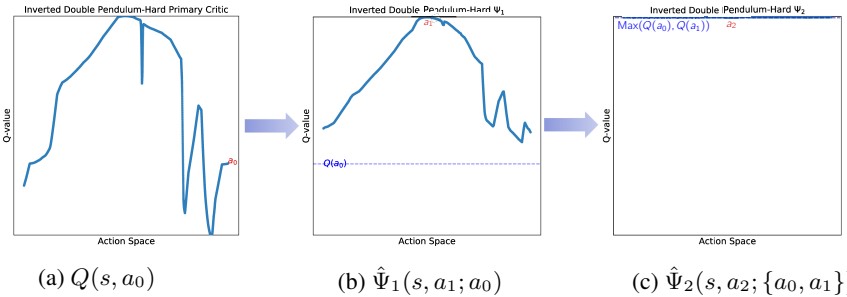

(a) $Q(s, a_0)$      (b) $\hat{\Psi}_1(s, a_1; a_0)$      (c) $\hat{\Psi}_2(s, a_2; \{a_0, a_1\})$

Figure 9: Each successive surrogate learns a Q-landscape with fewer local optima and thus is easier to optimize by its actor. SAVO helps a single actor escape the local optimum $a_0$ in Inverted Pendulum.

sampled actions. Appendix Figure 22 ablates SAVO in all 7 environments and shows that the most critical features are its successive nature, removing policy smoothing, and approximate surrogates.

## 6.2 Q-LANDSCAPE ANALYSIS: DO SUCCESSIVE SURROGATES REDUCE LOCAL OPTIMA?

In Figure 9, we visualize the surrogate landscapes in Inverted Pendulum-Restricted for one state $s$. Due to successive pruning and approximation, the Q-landscapes become smoother with reduced local optima. A single actor gets stuck in a severe local optimum $a_0$. However, surrogate $\Psi_1$ utilizes $a_0$ as an anchor and finds a better (global) optimum $a_1$. The maximizer policy selects $a_0, a_1$, or $a_2$, whichever has the highest Q-value. Appendix Figure 29 shows that convex Q-landscapes are easily optimized, while Figure 30 shows how SAVO successfully optimizes the non-convex Q-landscapes in all other tasks. Further analysis can be found in Appendix K.2.

## 6.3 CHALLENGING DEXTEROUS MANIPULATION (ADROIT)

In Adroit (Rajeswaran et al., 2017) dexterous manipulation on Door, Pen, and Hammer, we compared SAVO to TD3 (Fujimoto et al., 2018) and observed that SAVO successfully addressed the Q-landscape challenges in TD3 algorithm (Figure 10) and TD3 has been improved with SAVO.

## 6.4 QUANTITATIVE ANALYSIS: THE EFFECT OF SUCCESSIVE ACTORS AND SURROGATES

We investigate the effect of increasing the number of successive actor-surrogates in SAVO in Pendulum (Figure 11a) and MineWorld (Figure 11b). Additional actor-surrogates significantly help to reduce severe local optima initially. However, the improvement saturates as the suboptimality gap reduces. While main SAVO results use 3 actors, SAVO significantly improves with 10 actors (Figure 26, Figure 27) across tasks. We suggest a heuristic strategy to decide the number of actors in Section I.

Next, we show that successive actors are needed because a single actor can still get stuck in local optima, even with an optimal Q-function. In Figure 11c, we consider a SAVO agent trained to

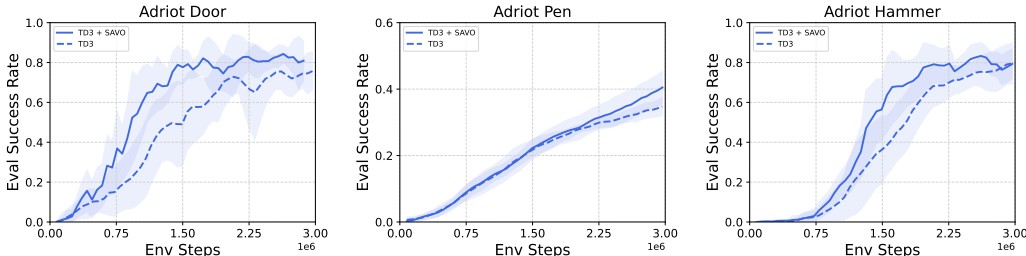

Figure 10: TD3 is improved with SAVO on Adroit dexterous manipulation tasks.

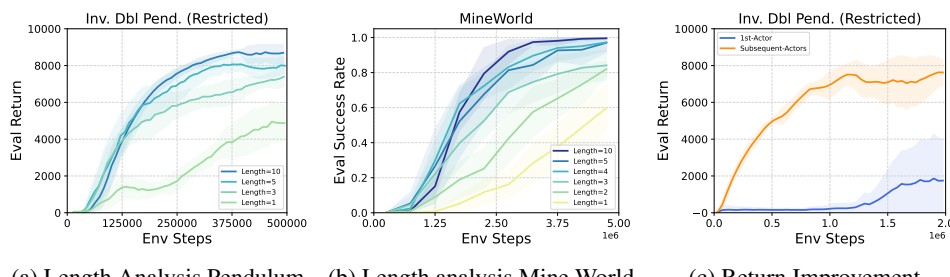

(a) Length Analysis Pendulum  (b) Length analysis Mine World  (c) Return Improvement

Figure 11: (L) More successive actor-surrogates are better, (R) SAVO v/s single-actor on inference.

optimality with 3 actors. When we remove the additional actors, the remaining single-actor agent resembles TD3 trained to maximize an "optimal" Q-function. However, the significant performance gap indicates that the single actor could not find optimal actions for the given Q-function.

## 6.5 FURTHER EXPERIMENTS TO VALIDATE SAVO

- **Baseline Optimization.** Figure 15 shows that baselines are fairly optimized, on par with SAVO on tasks with a simple Q-landscape. Hyperparameter optimization details are discussed in Section H.3.
- **SAVO orthogonal to SAC.** Figure 18 shows that SAVO+TD3 > SAC > TD3; thus, SAC's stochastic policy does not address TD3's non-convexity. In fact, SAC also suffers from local optima (Section J, Figure 28) that SAVO+SAC mitigates successfully in unrestricted Ant-v4 and Half-Cheetah-v4.
- **Design Choices.** Figure 20 shows that LSTM, DeepSet, and Transformers are all valid choices as summarizers of successive actions $a_{<i}$ in SAVO. Figure 21 shows that FiLM conditioning on $a_{<i}$ helps in discrete action spaces, but affects continuous action space less. For exploration, we compared Ornstein-Uhlenbeck (OU) noise and Gaussian noise and found them to be largely equivalent across all baselines (Figure 17). In Section G.7, we tried specialized initializations to enforce diversity in the SAVO's actors and surrogates but did not observe major gains.
- **Massive Discrete Actions.** SAVO also improves in RecSim-100k and RecSim-500k (Figure 19).
- **Resetting baselines.** SAVO outperforms resetting techniques (Nikishin et al., 2022; Kim et al., 2024) in addressing local optima, as shown in Figure 16.

## 7 LIMITATIONS AND CONCLUSION

Introducing more actors in SAVO has negligible influence on GPU memory, but leads to longer inference time (Table 1). However, even for 3 actor-surrogates, SAVO achieves significant improvements in all our experiments. Further,

| Method | GPU Mem. | Return | Time |
|---|---|---|---|
| TD3 | 619MB | 1107.795 | 0.062s |
| SAVO k=3 | 640MB | 2927.149 | 0.088s |
| SAVO k=5 | 681MB | 3517.319 | 0.122s |

Table 1: Compute v/s Performance Gain

for tasks with a simple convex Q-landscape, single actors do not get stuck in local optima, reducing the improvements with SAVO. In conclusion, we improve Q-landscape optimization in actor-critic RL with Successive Actors for Value Optimization (SAVO) in both continuous and large discrete action spaces. We demonstrate with quantitative and qualitative analyses how the improved optimization of Q-landscape with SAVO leads to better sample efficiency and performance.

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

APPENDIX

# Table of Contents

## A    REPRODUCIBILITY

With the aim of promising the reproducibility of our results, We have also included all relevant hyperparameters and additional details on how we tuned each baseline method in Appendix Table 3.

## B    PROOF OF CONVERGENCE OF MAXIMIZER ACTOR IN TABULAR SETTINGS

**Theorem B.1** (Convergence of Policy Iteration with Maximizer Actor). *Consider a modified policy iteration algorithm where, at each iteration, we have a set of $k + 1$ policies $\{\nu_0, \nu_1, \ldots, \nu_k\}$, with $\nu_0 = \mu$ being the current policy learned with DPG. We define the* maximizer actor $\mu_M$ *as:*

$$\mu_M(s) = \arg \max_{a \in \{\nu_0(s), \nu_1(s), \ldots, \nu_k(s)\}} Q(s, a). \tag{10}$$

*In the tabular setting, the modified policy iteration algorithm using the maximizer actor converges to the optimal policy.*

*Proof.* B.1   POLICY ITERATION WITH MAXIMIZER ACTOR

### B.2    POLICY EVALUATION CONVERGES

Given a deterministic policy $\pi$ (in our case $\pi = \mu_M$), the policy evaluation computes the action-value function $Q^\pi$, which satisfies the Bellman equation:

$$Q^\pi(s, a) = R(s, a) + \gamma \sum_{s'} P(s, a, s') Q^\pi(s', \pi(s')).$$

In the tabular setting, the Bellman operator $\mathcal{T}^\pi$ defined by

$$[\mathcal{T}^\pi Q](s, a) = R(s, a) + \gamma \sum_{s'} P(s, a, s') Q(s', \pi(s'))$$

is a contraction mapping with respect to the max norm $\| \cdot \|_\infty$ with contraction factor $\gamma$:

$$\|\mathcal{T}^\pi Q - \mathcal{T}^\pi Q'\|_\infty \leq \gamma \|Q - Q'\|_\infty.$$

Thus, iteratively applying $\mathcal{T}^\pi$ starting from any initial $Q_0$ converges to the unique fixed point $Q^\pi$.

### B.3    POLICY IMPROVEMENT WITH DPG AND MAXIMIZER ACTOR

At iteration $n$, suppose we have a policy $\mu_n$.

**Step 1: Policy Evaluation**

Compute $Q^{\mu_n}$ by solving:

$$Q^{\mu_n}(s, a) = R(s, a) + \gamma \sum_{s'} P(s, a, s') Q^{\mu_n}(s', \mu_n(s')).$$

**Step 2: Policy Improvement**

(a) *DPG Update*

Perform a gradient ascent step using the Deep Policy Gradient (DPG) method to obtain an improved policy $\tilde{\mu}_{k+1}$:

$$\tilde{\mu}_{k+1}(s) = \mu_n(s) + \alpha \nabla_a Q^{\mu_n}(s, a)\big|_{a=\mu_n(s)},$$

where $\alpha > 0$ is a suitable step size.

This DPG gradient step leads to local policy improvement following over $\mu_n$ (Silver et al., 2014):

$$V^{\tilde{\mu}_{k+1}}(s) \geq V^{\mu_n}(s), \quad \forall s \in \mathcal{S}.$$

(b) *Maximizer Actor*

Given additional policies $\nu_1, \ldots, \nu_k$, define the maximizer actor $\mu_{n+1}$ as:

$$\mu_{n+1}(s) = \arg \max_{a \in \{\tilde{\mu}_{k+1}(s), \nu_1(s), \ldots, \nu_k(s)\}} Q^{\mu_n}(s, a).$$

Since $\mu_{n+1}(s)$ selects the action maximizing $Q^{\mu_n}(s, a)$ among candidates, we have:

$$Q^{\mu_n}(s, \mu_{n+1}(s)) = \max_a Q^{\mu_n}(s, a) \geq Q^{\mu_n}(s, \tilde{\mu}_{k+1}(s)) \geq V^{\mu_n}(s).$$

By the Policy Improvement Theorem, since $Q^{\mu_n}(s, \mu_{n+1}(s)) \geq V^{\mu_n}(s)$ for all $s$, it follows that:

$$V^{\mu_{n+1}}(s) \geq V^{\mu_n}(s), \quad \forall s \in \mathcal{S}.$$

Thus, the sequence $\{V^{\mu_n}\}$ is monotonically non-decreasing.

### CONVERGENCE OF POLICY ITERATION

Since $\{V^{\mu_n}\}$ is bounded above by $V^*$ (the optimal value function), it converges. In a finite MDP, there are only finitely many possible policies. Thus, the sequence $\{\mu_n\}$ must eventually repeat, and because each policy improvement is non-decreasing, the policies stabilize at an optimal policy $\mu^*$.

$\square$

## C    PROOF OF REDUCING NUMBER OF LOCAL OPTIMA IN SUCCESSIVE SURROGATES

**Theorem C.1.** *Consider a state $s \in \mathcal{S}$, $Q$ in Eq. 1, and $\Psi_i$ in Eq. 7. Let $N_{opt}(f)$ be the number of local optima (assumed countable) of a function $f : \mathcal{A} \to \mathbb{R}$, where $\mathcal{A}$ is the action space. Then,*

$$N_{opt}(Q(s, a)) \geq N_{opt}(\Psi_0(s, a; \{a_0\})), \ldots, \geq N_{opt}(\Psi_k(s, a; \{a_0, \ldots, a_k\}))$$

*Proof.* Consider two consecutive surrogate functions $\Psi_i(s, a; \{a_0, \ldots, a_i\})$ and $\Psi_{i+1}(s, a; \{a_0, \ldots, a_{i+1}\})$,

$$\Psi_i(s, a; a_{<i}) = \max \left\{ Q(s, a), \max_{j<i} Q(s, a_j) \right\},$$

$$\Psi_{i+1}(s, a; a_{<i+1}) = \max \left\{ Q(s, a), \max_{j<i+1} Q(s, a_j) \right\},$$

Let $\tau_i = \max_{j<i} Q(s, a_j)$ and $\tau_{i+1} = \max_{j<i+1} Q(s, a_j)$.

Consider a given state $s$ and any particular local optimum in $\Psi_i$ at $a'$, there can be two cases:

1. If Q(s, a') > $\tau_{i+1}$, then $\Psi_{i+1}(s, a'; a_{<i+1}) = Q(s, a')$.

   Since, a' is a local optimum of $\Psi_i$, there exists $\epsilon > 0$ $\Psi_i(s, a' \pm \epsilon; a_{<i}) = Q(s, a' \pm \epsilon) < \Psi_i(s, a'; a_{<i}) = Q(s, a')$

   Therefore, $\Psi_{i+1}(s, a' \pm \epsilon; a_{<i+1}) = Q(s, a' \pm \epsilon) < \Psi_{i+1}(s, a'; a_{<i+1}) = Q(s, a')$ Thus, a' is also a local optimum of $\Psi_{i+1}$.

2. If $Q(s, a') \leq \tau_{i+1}$, then $\Psi_{i+1}(s, a'; a_{<i+1}) = \tau_{i+1}$, and there exists $\epsilon > 0$, such that $\Psi_{i+1}(s, a' \pm \epsilon; a_{<i+1}) = \tau_{i+1}$. Thus, a' is *not* a local optimum of $\Psi_{i+1}$

Finally, $\Psi_{i+1}$ does not add any new local optima, because $\tau_{i+1} \geq \tau_i$ and thus all points where $\Psi_{i+1}(s, a; a_{<i+1}) = Q(s, a)$, we have $\Psi_i(s, a; a_{<i}) = Q(s, a)$. Therefore $\forall i \geq 1$,

$$N_{\text{opt}}(\Psi_i(s, a; \{a_0, \ldots, a_i\})) \geq N_{\text{opt}}(\Psi_{i+1}(s, a; \{a_0, \ldots, a_{i+1}\}))$$

The same analysis extends for $Q$ and $\Psi_1$, by substituting $\tau_0 < \min Q$ to be a very small value. Thus, by induction, we have,

$$N_{\text{opt}}(Q(s,a)) \geq N_{\text{opt}}(\Psi_0(s,a; \{a_0\})), \ldots, \geq N_{\text{opt}}(\Psi_k(s,a; \{a_0, \ldots, a_k\}))$$

$\square$

## D  ENVIRONMENT DETAILS

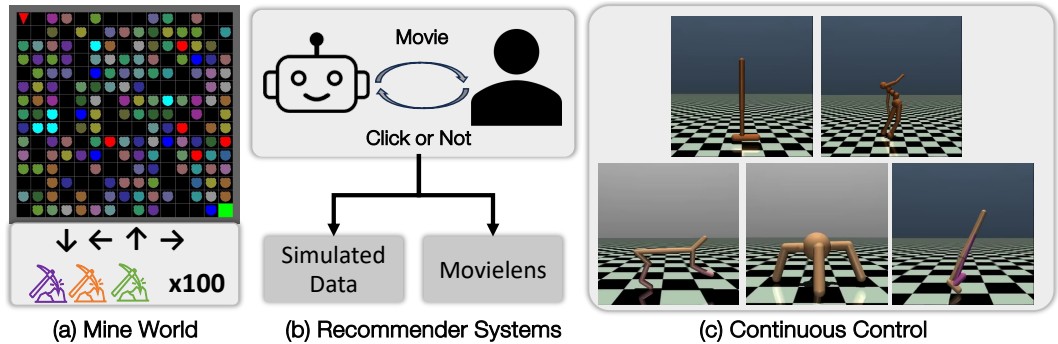

(a) Mine World          (b) Recommender Systems          (c) Continuous Control

Figure 12: This figure provides the visual description of the environment setup.

### D.1  MININGENV

The grid world environment, introduced in Sec. 5, requires an agent to reach a goal by navigating a 2D maze as soon as possible while breaking the mines blocking the way.

**State**: The state space is an 8+K dimensional vector, where K equals to *mine-category-size*. This vector consists of 4 independent pieces of information: Agent Position, Agent Direction, Surrounding Path, and Front Cell Type.

1. Agent Position: Agent Position occupies two dimensional of the vector. The first dimension represents the x-axis value, and the second one represents the y.

2. Agent Direction: It only takes one channel with value [0, 1, 2, 3]. Each number represents one direction, and they are 0-right, 1-down, 2-left, and 3-up.

3. Surrounding Path: This information takes four channels. Each represents whether the cell in that direction is an empty cell or a goal.

4. Front Cell Type: This information is in one-hot form and occupies the last K + 1-dimensional vector, which provides the information of which kind of mine is in front of the agent. If the front cell is an empty cell or the goal, the $(K+1)^{th}$ channel will be one, and others remain to be zero

Ultimately, we will normalize each dimension to [0, 1] with each dimension's minimum/maximum value. Each time we reset the environment, the layout of the whole grid world will be changed, except for the agent start position and the goal position.

**Termination**: An episode is terminated in success when the agent reaches the goal or after a total of 100 timesteps.

**Actions**: The base action set combines two kinds of actions: navigation actions and pick-axe(tool) actions. The navigation action set is a fixed set, which contains four independent actions: going up, down, left, and right, corresponding with the direction of the agent. They will change the agent's direction first and then try to make the agent take one step forward. Note that, different from the empty cell, the agent cannot step onto the mine, which means that if the agent is trying to take a step towards a mine or the border of the world, then the agent will stay in the same location while the direction will still be changed. Otherwise, the agent can step onto that cell. An agent will succeed if it reaches the goal position. The size of the pick-axe action set is equal to 50. Each tool has a

one-to-one mapping, which means they can and only can be successfully applied to one kind of mine, and either transform that kind of mine into another type of mine or directly break it.

**Reward**: The agent receives a large goal reward for reaching the goal. The goal reward is discounted based on the number of action steps taken to reach that location, thus rewarding shorter paths. To further encourage the agent to reach the goal, a small exploration reward is added whenever the agent gets closer to the goal, and a negative equal penalty is added whenever the agent gets further to the goal. And also, when the agent successfully applies a tool, it will gain a small reward. When the agent successfully breaks a mine, it will also gain a small bonus.

$$
\begin{aligned}
R(s, a) = &\ \mathbb{1}_{Goal} \cdot R_{\text{Goal}} \left( 1 - \lambda_{\text{Goal}} \frac{N_{\text{current steps}}}{N_{\text{max steps}}} \right) + \\
&\ R_{\text{Step}} \left( D_{\text{distance before}} - D_{\text{distance after}} \right) + \\
&\ \mathbb{1}_{correct\ tool\ applied} \cdot R_{\text{Tool}} + \\
&\ \mathbb{1}_{successfully\ break\ mine} \cdot R_{\text{Bonus}}
\end{aligned}
\tag{11}
$$

where $R_{\text{Goal}} = 10$, $R_{\text{Step}} = 0.1$, $R_{\text{Tool}} = 0.1$, $R_{\text{Bonus}} = 0.1$, $\lambda_{\text{Goal}} = 0.9$, $N_{\text{max steps}} = 100$

**Action Representations**: The action representations are 4-dimensional vectors manually defined using a mix of number ids, and each dim is scaled to [0, 1]. as shown in Graph 13. Dimensions 1 identifies the category of skills (navigation, pick-axe), 2 distinguishes movement skills (right, down, left, up), 3 denotes the mine on which this tool can be successfully applied, and 4 shows the result of applying this tool. We will normalize the action embedding space to [0, 1] for each dimension.

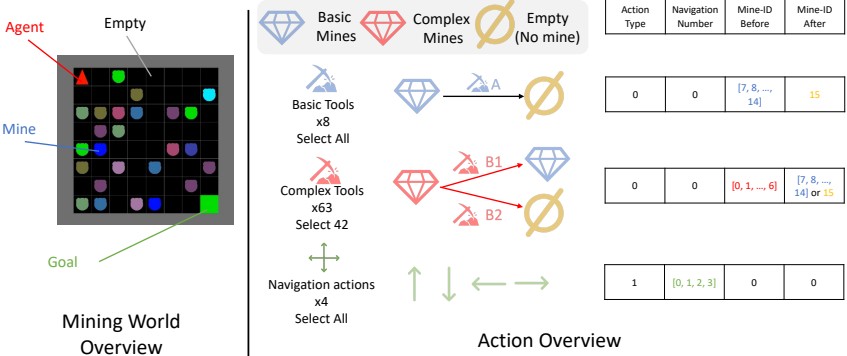

Figure 13: Mining Env Setting Description

## D.2 RECSIM

The simulated RecSys environment requires an agent to select an item that match the user's interest out of a large item-set. We simulate users with a dynamically changing preference upon clicks. Thus, the agent's task is to infer this preference from user clicks and recommend the most relevant item to maximize a total number of clicks.

**State**: The user interest embedding ($e_u \in \mathbb{R}^n$ where $n$ denotes the number of categories of items) represents the user interest in categories that transitions over time as the user consumes different items upon click. So, when the user clicks an item with the corresponding item embedding($e_i \in \mathbb{R}^n$; the same $n$ as the one for the user embedding) then the user interest embedding($e_u$) will be updated as follows;

$$
\begin{aligned}
\Delta(e_u) &= (-y|e_u| + y) \cdot (1 - e_u), \text{ for } y \in [0, 1] \\
e_i &\leftarrow e_u + \Delta(e_u) \text{ with probability}[e_u^T e_i + 1]/2 \\
e_u &\leftarrow e_u - \Delta(e_u) \text{ with probability}[1 - e_u^T e_i]/2
\end{aligned}
$$

This essentially pulls the user's preference towards the item that was clicked.

**Action**: The action set contains many recommendable items. So, the agent has to find the most relevant item to a user given the item-set. See below regarding how these items are represented.

**Reward**: The base reward is a simulated user feedback (e.g., clicks). The user model (Ie et al., 2019) stochastically skips or clicks the recommended item based on the present user interest embedding ($e_u$). Concretely, the user model computes the following score on the recommended item;

$$\text{score}_{item} = \langle e_u, e_i \rangle$$
$$p_{item} = \frac{e^{score_{item}}}{e^{s_{item}} + e^{score_{skip}}}$$
$$p_{skip} = \frac{e^{score_{skip}}}{e^{s_{item}} + e^{score_{skip}}}$$

where, $e_u, e_i \in \mathbb{R}^n$ are the user and item embedding, respectively, $\langle \cdot, \cdot \rangle$ is the dot product notation and $score_{skip}$ is a empirically decided hyper-parameter. So, given the score $score_{item}$ of an item, the user model computes the click likelihood through a softmax function over the recommended item and a predefined skip score. Finally, the user model stochastically selects either click(reward=1) or skip(reward=0) based on the categorical distribution on $[p_{item}, p_{skip}]$.

**Action Representations**: Following Jain et al. (2021), we implement continuous item representations sampled from a Gaussian Mixture Model (GMM) with centers around each item category. In this work, we did not use the sub-category in the category system.

### D.3 CONTINUOUS CONTROL

The MuJoCo (Todorov et al., 2012) benchmarking tasks are a set of standard reinforcement learning environments provided by the MuJoCo physics engine. elow is a brief description of some of the commonly used MuJoCo benchmarking tasks:

**Hopper**: In the Hopper task, you control a one-legged robot that must learn to hop forward while maintaining balance. The agent needs to find an optimal hopping strategy to maximize forward progress.

**Walker2d**: This task features a two-legged robot that must learn to walk forward. Similar to the Hopper, the agent must maintain balance while moving efficiently.

**HalfCheetah**: The HalfCheetah task involves a four-legged cheetah-like robot. The objective is for the robot to learn a coordinated gait that allows it to move forward as rapidly as possible.

**Ant**: In the Ant task, you control a four-legged ant-like robot. The challenge is for the robot to learn to walk and navigate efficiently through its environment.

#### D.3.1 RESTRICTED LOCOMOTION IN MUJOCO

Figure 6 demonstrates "Restricted" locomotion. And here we provide the complete description and justification of the Restricted Mujoco Locomotion tasks below.

**Justification**: The restricted locomotion setting in Mujoco limits the range of actions the agent is allowed to perform in each dimension. For instance, the wear and tear of an agent's hardware can easily cause action space to behave like the one visualized in the attached PDF for Hopper. The mixture-of-hypersphere action space is just one way to simulate such asymmetric restrictions. These restrictions apply to the range of torques applied to the joints of hopper and walker, and on the range of forces applied to pendulums.

**Complete Description**:

- **Restricted Hopper & Walker**

  Invalid action vectors are replaced with 0. Change to environment's step function code:

```
1  def step(action):
2      ...
3      if check_valid(action):
```

```
4          self.do_simulation(action)
5      else:
6          self.do_simulation(np.zeros_like(action))
7      ...
```

For reference, the Hopper action space is 3-dimensional, with torque applied to [thigh, leg, foot], while the Walker action space is 6-dimensional, with torque applied to [right thigh, right leg, right foot, left thigh, left leg, left foot]. The implication is that zero torques are exerted for the $\Delta t$ duration between two actions, meaning no torques are applied for $0.008$ seconds. This effectively slows down the agent's current velocities and angular velocities due to friction.

- **Inverted Pendulum & Inverted Double Pendulum**

Invalid action vectors are replaced with -1. Change in code:

```
1  def step(action):
2  ...
3      if not check_valid(action):
4          action[:] = -1.
5      self.do_simulation(action)
6  ...
```

For reference, the action space is 1-dimensional, with force applied on the cart. The implication is that the cart is pushed in the left direction for $0.02$ (default) seconds. Note that the action vectors are not zeroed because a $0$-action is often the optimal action, particularly when the agent starts upright. This would make the learning task trivial, with the optimal strategy being: "learn to select invalid actions".

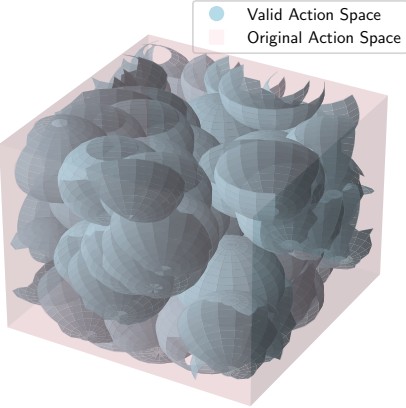

Figure 14: Hopper's 3D visualization of Action Space.

# E  ADDITIONAL RESULTS

## E.1  EXPERIMENT: CONTINUOUS CONTROL ON UNRESTRICTED MUJOCO

In Mujoco-v4 tasks, the Q-landscape is likely to be easier to optimize than Mujoco-Restricted tasks, and we find that baseline models consistently perform well in all the tasks, unlike Mujoco-Restricted. Based on the performance of SAVO and baselines in Figure 15, we can infer that,

1. The baseline models have sufficient capacity and are well-tuned, as they can solve the standard Mujoco-v4 tasks optimally.

2. SAVO performs on par with other methods in Mujoco-v4 tasks where the Q-landscape is easier to optimize.

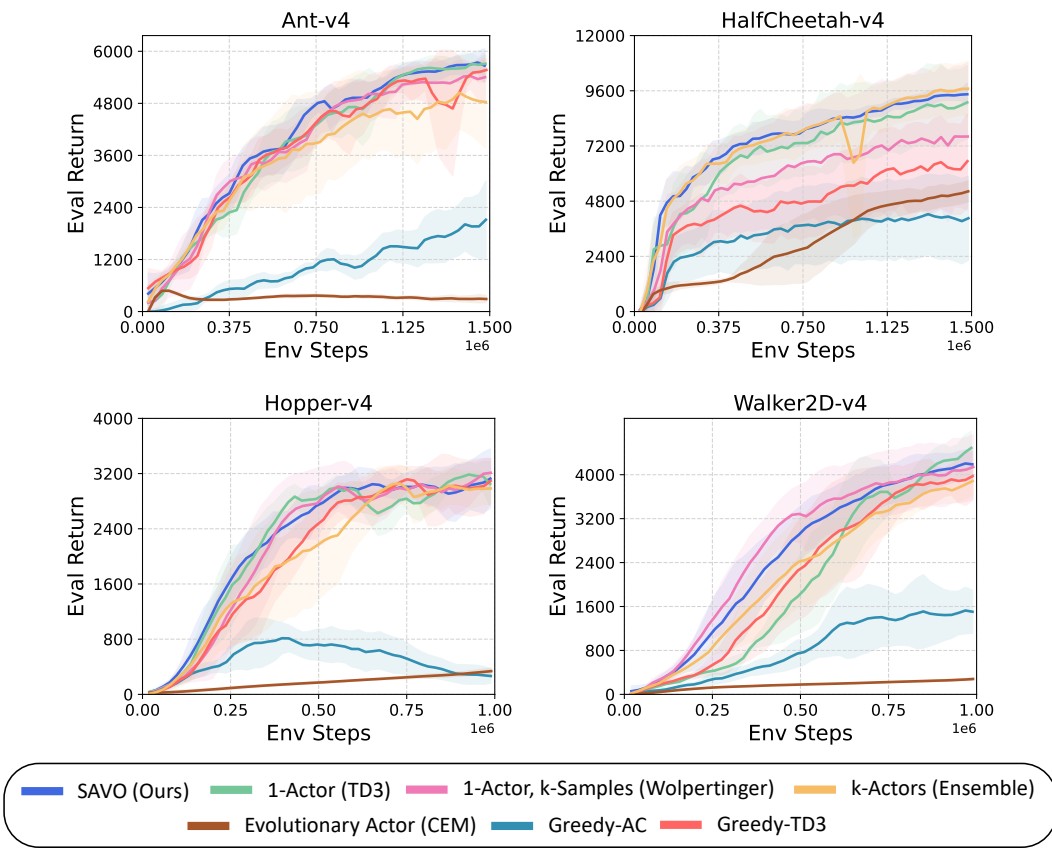

Figure 15: **TD3 is not suboptimal in Unrestricted Mujoco**. We evaluate SAVO against all baselines in the Unrestricted Mujoco continuous control tasks and show that SAVO is competitive with the baselines that already perform optimally. The reason is investigated in Section K, where tasks like Inverted Pendulum-v4 and Hopper-v4 have visibly convex Q-landscapes. Thus, SAVO is not expected to significantly outperform TD3 in these benchmarks.

3. Since SAVO outperforms baseline methods only in Mujoco-Restricted, it demonstrates that the reason of SAVO doing better is the presence of a challenging Q-landscape, such as those shown in Figure 2.

### E.2 RESETTING BASELINES

In this section, we clarify the distinction between primacy bias and the challenge of getting stuck in local optima within Q-landscapes. Primacy bias, as addressed in Nikishin et al. (2022); Kim et al. (2024), occurs when an agent is trapped in suboptimal behaviors from early training, and solutions like resetting (reinitializing the parameters of last few layers) and re-learning from the replay buffer mitigate this by avoiding reliance on initially collected samples.

However, these methods do not reduce the probability of an actor getting stuck in Q-function local optima (the issue we consider in this work). In fact resetting could cause an otherwise optimal actor to get stuck in suboptima during retraining. To demonstrate this, we conducted a reset baseline experiment, following Nikishin et al. (2022), on TD3 in **MineEnv**. Here, *Full-reset* refers to the *reset all* strategy proposed by Kim et al. (2024), while *Last-layer-OOO* corresponds to the approach in Nikishin et al. (2022). Finally, *TD3 (no reset)* represents the standard TD3 algorithm without these extensions. We observed no performance improvements over the standard TD3. In contrast, our method, SAVO, directly addresses this problem by employing an actor architecture specifically designed to navigate non-convex Q-landscapes, making it more robust to local optima.

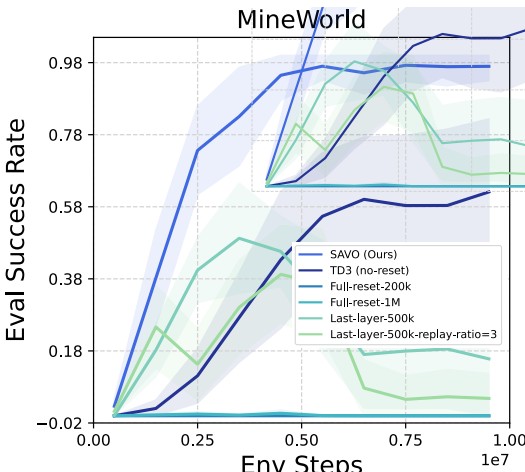

Figure 16: Performance comparisons of Resetting baselines averaged over 5 random seeds, and the seed variance is shown with shading.

### E.3 EXPLORATION NOISE COMPARISON: OUNOISE VS GAUSSIAN

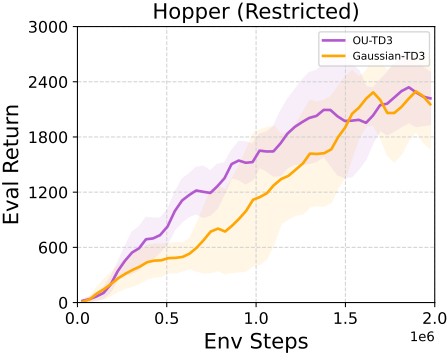

Figure 17: **OU versus Gaussian Noise**. We do not see a significant difference due to this choice, and select OU noise due to better overall performance in experiments

The choice of Ornstein-Uhlenbeck (OU) noise or Gaussian noise for exploration does not make a significant difference and we select OU noise for its better overall performance in initial experiments. This comparison is shown in Figure 17. This finding is consistent with TD3 Fujimoto et al. (2018), which also finds no significant difference between OU and Gaussian noise.

### E.4 SAC DOES NOT ADDRESS NON-CONVEX Q-LANDSCAPES

We compare the performance of SAC, TD3, and TD3 + SAVO across three Mujoco-Restricted tasks. The results (Figure 18) indicate that TD3 + SAVO consistently outperforms the other methods, demonstrating the effectiveness of SAVO in *Hopper* and *Walker2D*. In *Inverted Pendulum*, TD3 + SAVO also shows faster convergence, further highlighting its advantages.

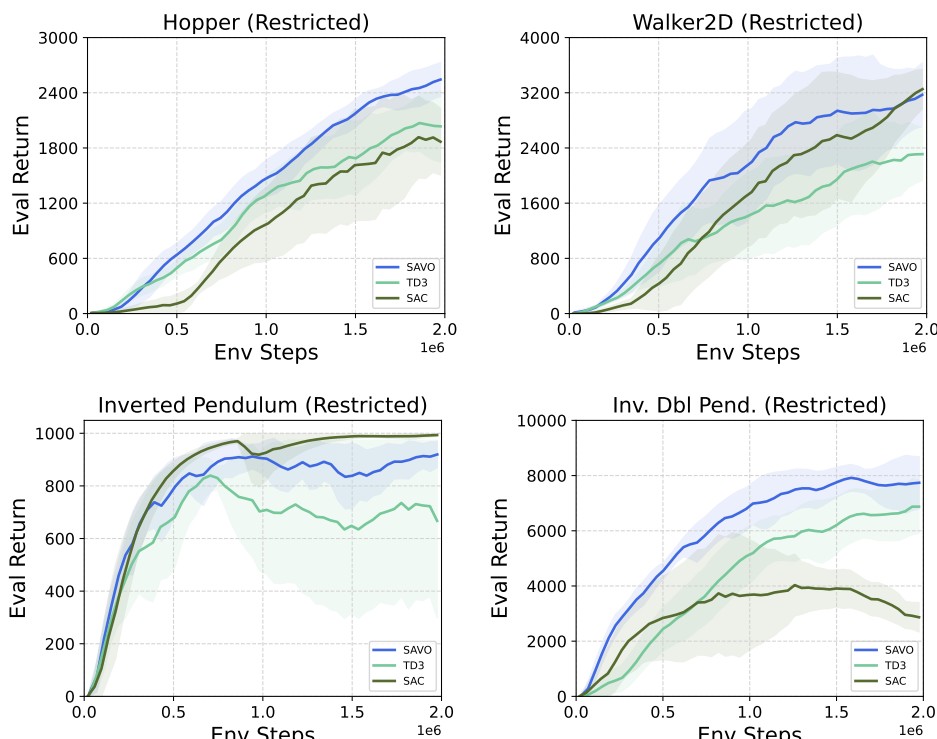

Figure 18: **SAC is orthogonal to the effect of SAVO**. While SAC is a stochastic extension of TD3 with entropy regularization, SAVO is a plug-in *actor architecture* that mitigates the challenge of the actor getting stuck in local optima. Thus, tasks where SAC outperforms TD3 differ from tasks where SAVO outperforms TD3. Also, TD3 outperforms SAC in Restricted Hopper and Inverted-Double-Pendulum. However, SAVO+TD3 guarantees improvement over TD3. As we show in Section J, SAVO+SAC also mitigates the local optima challenges in SAC.

# F    NETWORK ARCHITECTURES

## F.1    SUCCESSIVE ACTORS

The whole actor has a successive format and each successive actor will receive two pieces of information: the state observation and the action list generated by previous successive actors. Given the concatenation of the input components above, a 4-layer MLP with ReLU will process this information and generate one action for one single successive actor. And this action will be concatenated with the previous action list. After being transformed by an optional action-list-encoder, together with the state information, they become the input of next successive actor's input. In the end, the action list will be processed with 1-NN to find the nearest discrete action. After this, this action list will be delivered to the selection Q-network.

## F.2    SUCCESSIVE CRITICS

The critic has a one-to-one mapping relationship with the actor. The whole critic consists of a list of successive critics and each successive critic will receive three pieces of information: the state observation, the action list generated by previous successive actors, and the action provided by the corresponding successive actor. Given the concatenation of the input components above, a 2-layer MLP with ReLU will process this information and generate the action's value for one single successive actor. This value will be used to update itself and the actor with TD-error.

### F.3 LIST SUMMARIZERS

In order to extract meaningful information from the list of candidate actions, following Jain et al. (2021) we employed the sequential models and the list-summarizer as follows;

**Bi-LSTM**: The raw action representations of candidate actions are passed on to the 2-layer MLP followed by ReLU. Then, the output of the MLP is processed by a 2-layer bidirectional LSTM (Huang et al., 2015). Another 2-layer MLP follows this to create the action set summary to be used in the following successive actor.

**DeepSet**: The raw action representations of candidate actions are passed on to the 2-layer MLP followed by ReLU. Then, the output of the MLP is aggregated by the mean pooling over all the candidate actions to compress the information. Finally, the 2-layer MLP with ReLU provides the resultant action summary to the following successive actor.

**Transformer**: Similar to the Bi-LSTM variant of the summarizer, we employed the 2-layer MLP with ReLU before inputting the candidate actions into a self-attention and feed-forward network to summarize the information. Afterward the summarization will be part of the input of the following successive actor.

### F.4 FEATURE-WISE LINEAR MODULATION (FiLM)

Feature-wise Linear Modulation (Perez et al., 2018), is a technique commonly applied in neural networks for tasks like image recognition. It enhances adaptability by dynamically adjusting intermediate feature representations. Using learned parameters from one layer, FiLM linearly modulates features in another layer, allowing the network to selectively emphasize or de-emphasize aspects of the input data. This flexibility is beneficial for capturing complex and context-specific relationships, improving the model's performance in various tasks.

### F.5 SELECTION Q-NETWORK

The selection Q-network sequentially evaluates the Q-value of the retrieved candidate actions by the cascading actors. Thus, it receives a concatenated information of state and an action embedding for each candidate action. Then, it selects the action with the largest Q-value amongst candidate actions to act on the environment.

## G    MORE EXPERIMENTAL RESULTS

### G.1    MORE COMPLEX RECSIM: INCREASING SIZE OF ACTION SPACE

We test the robustness of our method to more challenging Q-value landscapes in Figure 19 in Appendix G.1. In RecSim, we vary the action space size, from $100K$ to $500K$. The results show that SAVO outperforms the baselines, maintaining its robust performance even as the action complexity increases. In contrast, the baselines experienced performance deterioration as action sizes grew larger.

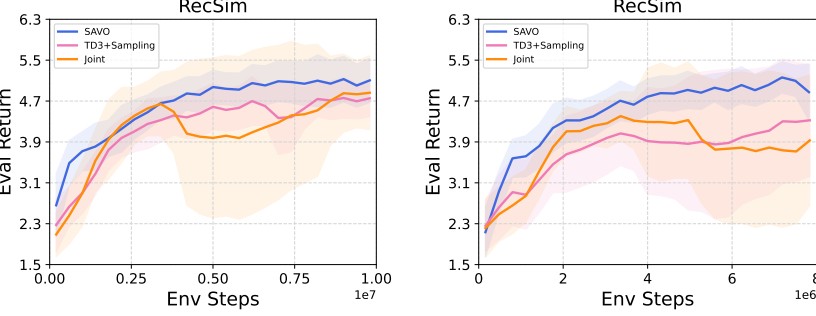

Figure 19: Increasing RecSim action set size: (Left) $100K$ items, (Right) $500K$ items (6 seeds).

## G.2 Design Choices: Action summarizers

In the exploration of action summarizer design choices, three key architectures were considered: Deepset, LSTM, and Transformer models, each represented by SAVO, SAVO-lstm, and SAVO-transformer in Fig.20, respectively. In the discrete tasks, the comparison revealed a preference for the deepset architecture over LSTM and Transformer. In the continuous domain, however, the results were rather varied, indicating that the effectiveness of the action summarizer depends on the specific use case. The nuanced differences among these architectures contribute to the complexity of the task, and further research is needed to determine the optimal design for action summarization in both discrete and continuous contexts.

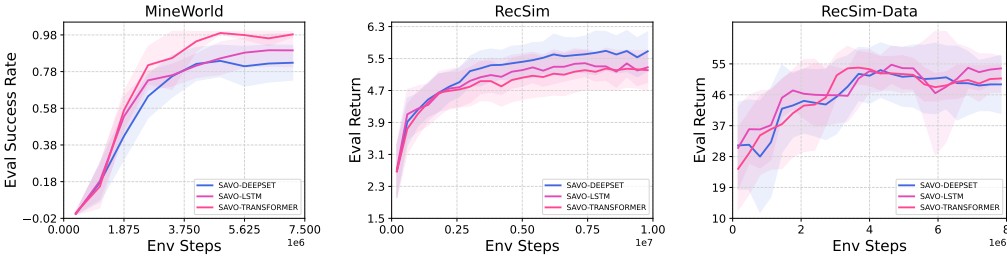

Figure 20: Comparison of action summarizers: the results are averaged over 5 random seeds, and the seed variance is shown with shading.

## G.3 Conditioning on previous actions: FiLM v/s MLP

In the examination of conditioning on previous actions, two distinct approaches, Feature-wise Linear Modulation (FiLM) and Multi-Layer Perceptron (MLP), represented by FiLM and non-FiLM variants in Fig.21, were scrutinized for their efficacy. In the discrete tasks, the results unveiled a notable preference for FiLM over non-FiLM implementations, highlighting its effectiveness in leveraging information from prior actions for improved conditioning. However, in the continuous domains, the comparison between FiLM and MLP yielded varied outcomes, suggesting that the choice between these approaches is intricately tied to the specific task context. The nuanced performance differences observed underscore the need for continued research to ascertain the optimal approach for conditioning on previous actions and to enhance model adaptability across diverse applications.

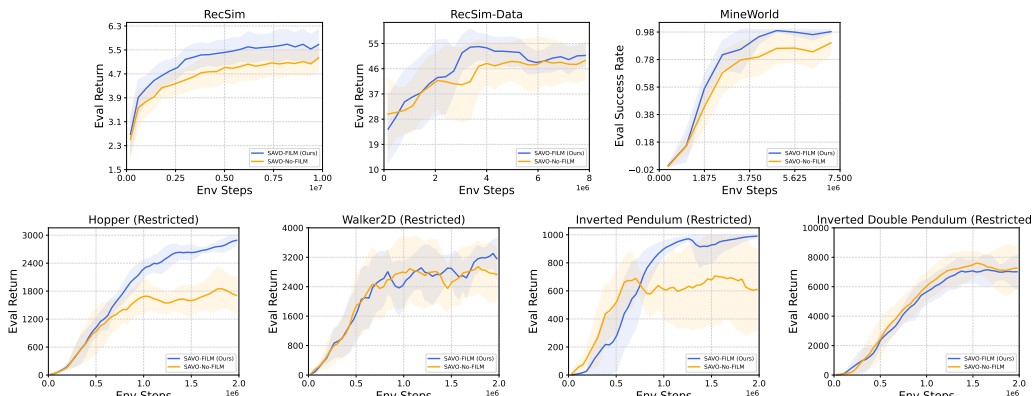

Figure 21: Comparison of how to condition on previous actions: the results are averaged over 5 random seeds, and the seed variance is shown with shading.

## G.4 Per-Environment Ablation Results

Figure 22 shows the per-environment performance of SAVO ablations, compiled into aggregate performance profiles in Figure 7b. The **SAVO - Approximation** variant underperforms significantly in discrete action space tasks, where traversing between local optima is complex due to nearby actions

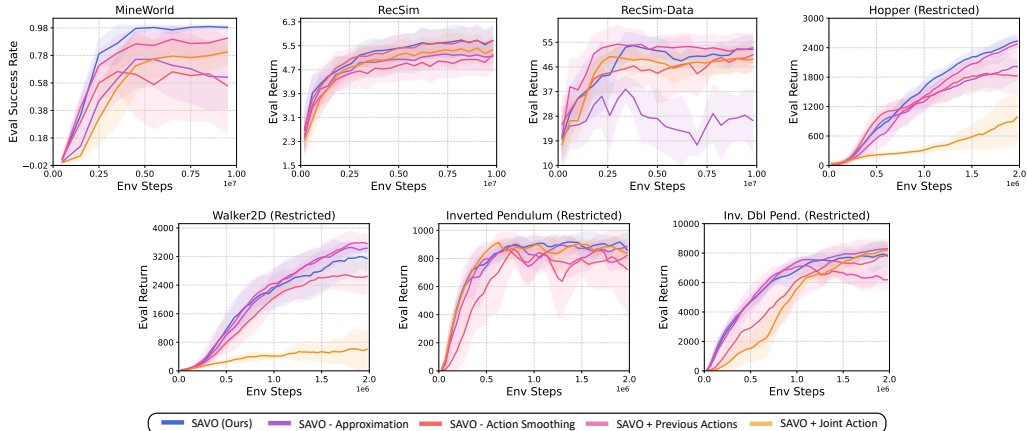

Figure 22: **Ablation study of SAVO Variations** over 5 random seeds shows that every technical component introduced in SAVO contributes to its performance.

having diverse Q-values (see the right panel of Figure 2). Similarly, adding TD3's target action smoothing to SAVO results in inaccurate learned Q-values when several differently valued actions exist near the target action, as in the complex landscapes of all tasks considered.

Removing information about preceding actions does not significantly degrade SAVO's performance since preceding actions' Q-values are indirectly incorporated into the surrogates' training objective (see Eq. 9), except for MineWorld where this information helps improve efficiency.

The **SAVO + Joint** ablation learns a single actor that outputs a joint action composed of $k$ constituents, aiming to cover the action space so that multiple coordinated actions can better maximize the Q-function compared to a single action. However, this increases the complexity of the architecture and only works in low-dimensional tasks like Inverted-Pendulum and Inverted-Double-Pendulum. SAVO simplifies action candidate generation by using several successive actors with specialized objectives, enabling easier training without exploding the action space.

## G.5  SURROGATE APPROXIMATION ERROR ANALYSIS

In Figure 23, we analyze the surrogate approximation error across different environments to evaluate how well the surrogate Q-functions approximate the true thresholded Q-function during training. The surrogate error, i.e., the MSE loss from Equation 9, is expressed as a percentage of the Bellman error to measure how closely the surrogate tracks updates to the Q-function. This analysis is important because surrogates aim to simplify optimization while still allowing gradients to propagate effectively.

**Low Surrogate Error Across Training.** In most environments, the surrogate error converges to a relatively low value between 1–10% of the Bellman error, showing that the surrogates provide a reliable approximation. This indicates that the surrogate functions are well-suited for actor updates, not introducing large errors in the Q-landscape and staying current with new optimal regions. The surrogate error stays consistently low across various tasks, including restricted locomotion (e.g., Hopper, Walker2D) and dexterous manipulation (e.g., Adroit Pen, Adroit Hammer). This demonstrates that the surrogate functions work well across diverse environments with varying levels of complexity.

**Non-zero loss shows Smoothness in Flat Regions.** The surrogate error remains positive throughout training, including in flat regions of the Q-landscape. This ensures that gradients can still propagate, preventing the actor from getting stuck in areas without gradient information.

**Behavior in the Inverted Double Pendulum (Restricted).** For the *Inverted Double Pendulum (Restricted)* environment, the surrogate error increases towards the end of training. This happens because the agent has already converged, and the increase in error reflects overtraining, which is consistent with the observation of an unstable drop in task performance for certain seeds.

Overall, this analysis shows that surrogate functions effectively simplify the Q-value landscape, closely track Q-function updates, and maintain robustness across different tasks, justifying their effectiveness in enabling gradient flow in SAVO. This results in SAVO outperforming the SAVO - Approximation baseline, as shown in Figure 7b and Figure 22.

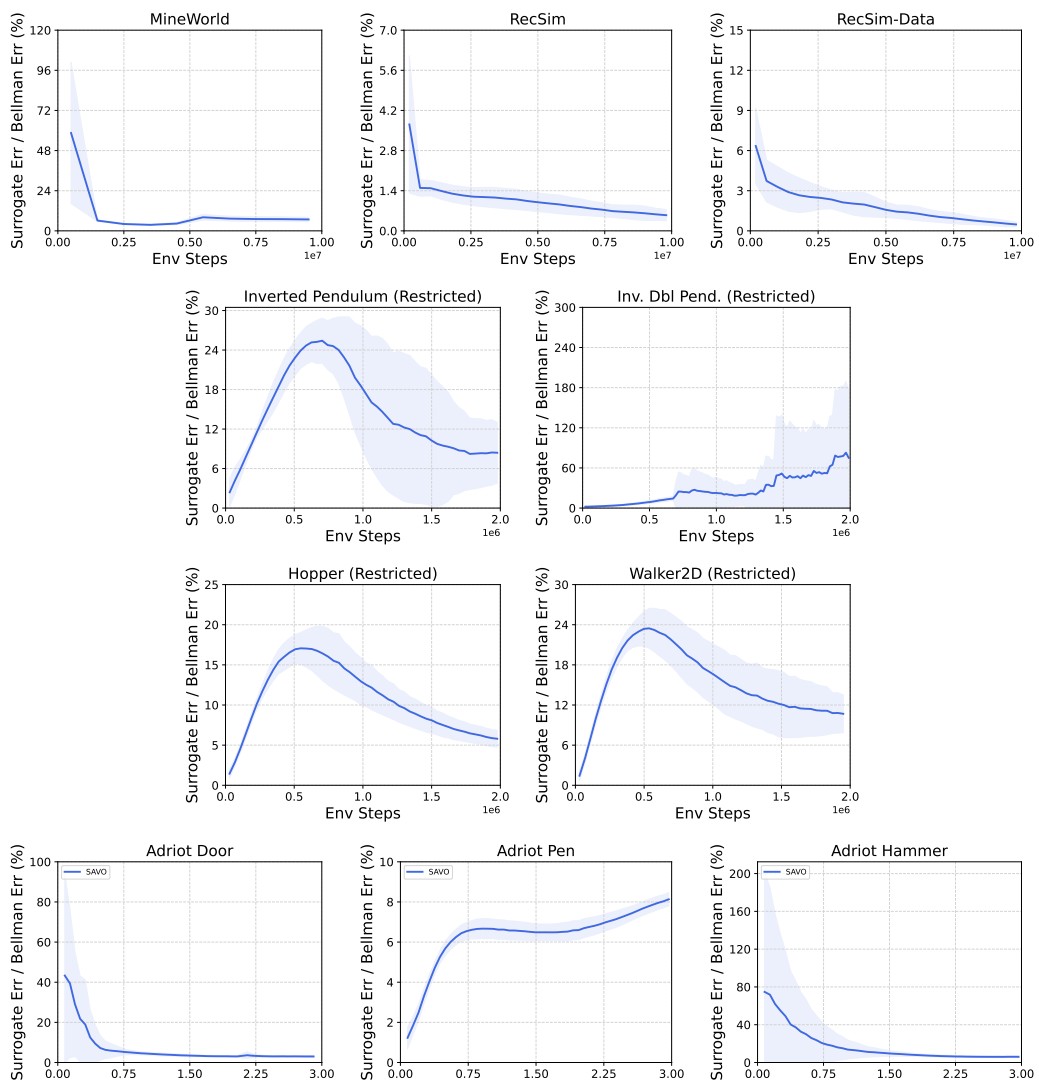

Figure 23: **Surrogate Approximation Error Analysis**. The plot shows the surrogate approximation error as a percentage of the Bellman error during training across various environments: $\frac{\text{Surrogate Approximation Error}}{\text{Bellman Error}}\%$. In most tasks, the surrogate loss converges to a relatively low value (within 1–10% of the Bellman error), indicating that (i) the surrogates effectively track updates to the Q-function, and (ii) the surrogate loss remains strictly positive, highlighting the smoothness of the surrogate landscape, especially in flat regions, where the exact approximation is undesirable to maintain effective gradient propagation. Notably, for the *Inverted Double Pendulum (Restricted)* environment, a rise in approximation error is observed towards the end of training. Upon further investigation, this was attributed to overtraining after the agent had already converged, corresponding to an unstable decline in task performance.

### G.6 Q-Smoothing Analysis: Discrete vs. Continuous Action Spaces

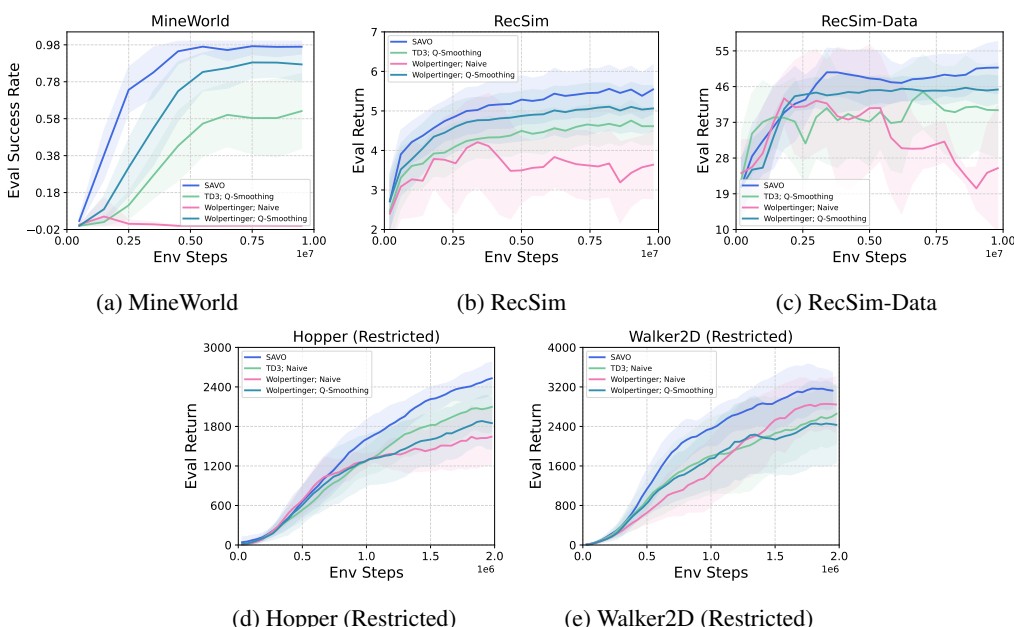

(a) MineWorld          (b) RecSim          (c) RecSim-Data

(d) Hopper (Restricted)          (e) Walker2D (Restricted)

Figure 24: **Impact of Q-smoothing**. The plots compare the performance of baselines with and without Q-smoothing. Results are averaged over 5 random seeds, with shading indicating variance. Q-smoothing benefits discrete tasks but has negligible impact in continuous action spaces.

| Baseline | MineWorld | RecSim | RecSim-Data | Hopper | Walker2D |
|---|---|---|---|---|---|
| TD3 | 0.6 | 4.5 | 42 | 2100 | 2700 |
| Wolpertinger [Naive] | 0.0 | 3.9 | 40 | 1650 | 2900 |
| Wolpertinger [Q-Smoothing] | 0.9 | 5.0 | 46 | 1850 | 2400 |
| SAVO (Ours) | 0.98 | 5.5 | 51 | 2500 | 3200 |

Table 2: **Q-smoothing in discrete tasks**. We compare the performance of baselines with and without Q-smoothing across tasks. Underline denotes which variant, naive or Q-smoothing, is used in the paper results. Wolpertinger [Naive] significantly underperforms in discrete action space tasks (denoted in red), and thus, we reported results on Wolpertinger [Q-Smoothing] in the paper. In continuous action space tasks, there was no benefit to Q-smoothing, and thus we chose to report results on Wolpertinger [Naive] as it is closer to the underlying TD3 algorithm. Note that the same Q-smoothing principle is applied for TD3 and SAVO, too, i.e., their Q-function is smoothed for better gradients in discrete action spaces, but unsmoothed Q-function is used in continuous action spaces.

The approximate surrogates introduced in Section 4.4 also have a smoothing effect on the Q-landscape that might ease gradient flow. A similar smoothing can be applied to the primary Q-function. We found such *Q-smoothing*, which involves learning an auxiliary Q-function to approximate and smooth the primary Q-function, to be essential for discrete action spaces. Q-smoothing facilitates the necessary gradient flow in discrete action space tasks because the primary Q-function is only trained on action representations corresponding to a finite number of discrete actions, while the intermediate action representations might have arbitrary values. By learning an approximate Q-function, the regions between the true action representations are smoothed, facilitating gradient flow.

Thus, in all baselines and SAVO in discrete action space tasks, we included Q-smoothing. However, we did not notice any benefit of Q-smoothing in continuous action space tasks, and thus, all baselines and SAVO do not have Q-smoothing. SAVO still has surrogate smoothing in all environments, because non-smoothed surrogates do not let gradient flow through flat regions.

To demonstrate the impact of Q-smoothing in both discrete and continuous action spaces, we conducted a detailed analysis across several tasks in Figure 24 and Table 2. This section investigates its efficacy and highlights the nuanced differences in its utility across environments.

**Discrete Action Spaces: Importance of Q-Smoothing.**

For discrete tasks, smoothing the Q-function significantly enhances performance by mitigating the complexity of local optima in diverse Q-value landscapes. This experiment primarily compares 1-Actor k-samples Wolpertinger-Naive and Wolpertinger-Q-smoothing approaches. As shown in Fig. 24, Q-smoothing is essential for Wolpertinger to perform well, while the non-smoothed counterparts significantly suffer in MineWorld and RecSim tasks. Note that the TD3 and SAVO results also *include* Q-smoothing.

**Continuous Action Spaces: Limited Impact of Q-Smoothing.**

In continuous action spaces, Q-smoothing does not yield a significant performance gain. In Wolpertinger, both the naive and Q-smoothing variants show comparable performance, indicating sufficient gradient information is present throughout the action space (unlike discrete action space tasks that have missing true Q-values).

For these tasks, as shown in Fig. 24, the introduction of Q-smoothing neither improves nor degrades performance. This justifies its exclusion from our continuous action space experiments and explains why we reported results for Wolpertinger [Naive] in these environments, as it is closer to the underlying TD3 algorithm. Note that the TD3 and SAVO results also *exclude* Q-smoothing.

**Conclusion.** Q-smoothing is crucial for discrete action space tasks, as demonstrated by its strong performance in our results. However, it provides no added value for continuous tasks. Consequently, our baselines reflect these observations, ensuring fair comparisons across all evaluated methods.

## G.7 SPECIALIZED INITIALIZATION STRATEGIES FOR DIVERSITY IN SAVO

To explore the potential impact of diverse policy and surrogate value function initializations on algorithm performance, we tested two specialized initialization strategies beyond the default Xavier initialization (Glorot & Bengio, 2010):

- **Xavier** (default). Weights are initialized with the default initialization: $w \sim$ Xavier-init

- **Random**. Weights are initialized from a standard normal distribution, i.e., $w \sim \mathcal{N}(0, 1)$.

- **Add**. Weights are initialized using Xavier initialization, followed by the addition of scaled standard normal noise, i.e., $x \sim$ Xavier-init, $y \sim 0.5 \cdot \mathcal{N}(0, 1)$, and $w = x + y$.

We compare these specialized initialization strategies in various tasks, with reward curves reported in Fig. 25 and summarized below:

- **MineWorld:** Add $\approx$ Random $\approx$ Xavier

- **RecSim:** Add $\approx$ Random $\approx$ Xavier

- **Hopper (Restricted):** Add $\approx$ Random $\approx$ Xavier

- **Adroit Door:** Add $\approx$ Random $<$ **Xavier**

**Findings.** The results indicate that specialized initialization strategies aimed at increasing diversity do not particularly improve performance. Across most tasks, Add and Random strategies perform similarly to standard Xavier initialization. However, in the Adroit Door task, the specialized initializations underperform compared to Xavier, suggesting that task-specific factors might influence the effectiveness of standard initialization strategies.

**Conclusion.** While our experiments show no significant benefit from specialized initialization strategies, the idea of explicitly incorporating diversity into the optimization process remains promising. We believe that designing algorithms with explicit diversity objectives *throughout training* could serve as a valuable heuristic in future work.

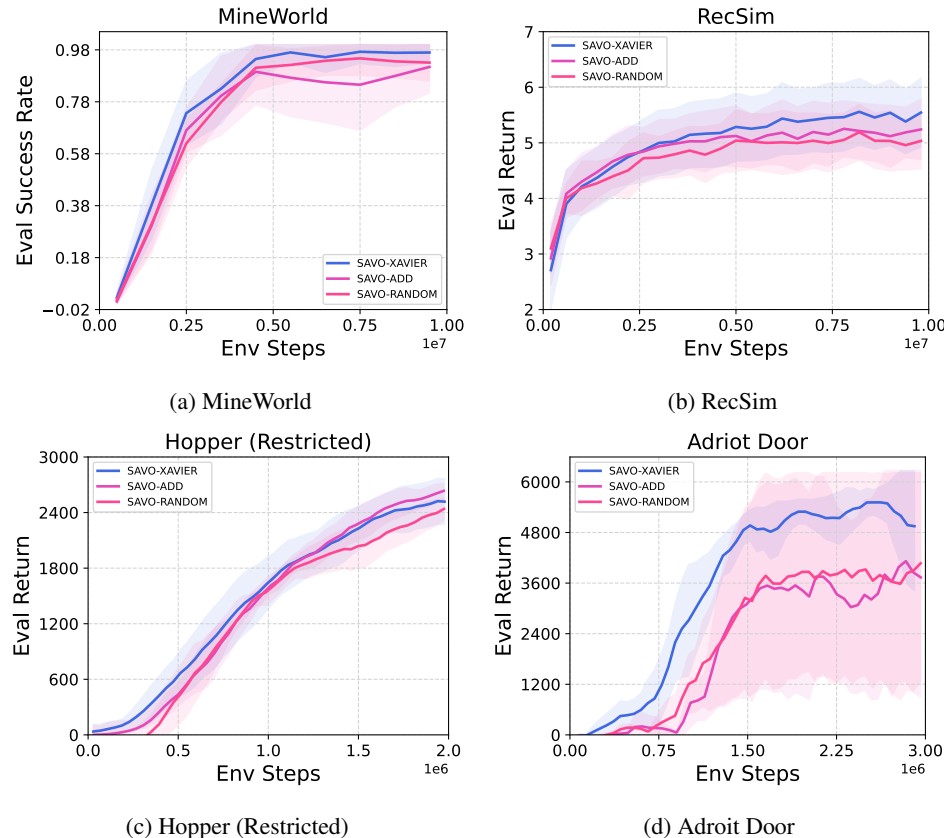

(a) MineWorld

(b) RecSim

(c) Hopper (Restricted)

(d) Adroit Door

Figure 25: **Specialized Initialization Strategies**. Reward curves compare Random and Add strategies to standard Xavier initialization across 4 tasks, showing no significant advantage of specialized initialization for increasing diversity.

# H EXPERIMENT DETAILS

## H.1 AGGREGATED RESULTS: PERFORMANCE PROFILES

Agarwal et al. (2021) proposed a robust means to rigorously validate the efficacy of our approach. Through the incorporation of the suggested performance profile, we have conducted a more thorough comparison of our approach against baselines, resulting in a comprehensive understanding of the inherent statistical uncertainty in our results. In Figure 7a, the x-axis illustrates normalized scores across all tasks, employing *min-max scaling* to normalize scores based on the initial performance of untrained agents aggregated across random seeds (i.e., *Min*) and the final performance presented in Figure 8 (i.e., *Max*).

Figure 7a reveals the consistent high performance of our method across various random seeds, with its curve consistently ranking at the top of the x-axis changes, while baseline curves exhibit earlier declines compared to our approach. This visual evidence substantiates the robustness and reliability of our method across different experimental conditions.

## H.2 IMPLEMENTATION DETAILS

We used PyTorch (Paszke et al., 2019) for our implementation, and the experiments were primarily conducted on workstations with either NVIDIA GeForce RTX 2080 Ti, P40, or V32 GPUs on. Each experiment seed takes about 4-6 hours for Mine World, 12-72 hours for Mujoco, and 6-72 hours for RecSim, to converge. We use the Weights & Biases tool (Biewald, 2020) for plotting and logging experiments. All the environments were developed using the OpenAI Gym Wrapper (Brockman et al., 2016). We use the Adam optimizer (Kingma & Ba, 2014) throughout.

## H.3 HYPERPARAMETERS

The environment-specific and RL algorithm hyperparameters are described in Table 3.

## H.4 COMMON HYPERPARAMETER TUNING

To ensure fairness across all baselines and our methods, We searched over hyper-parameters that are common across baselines;

- **Learning rate of Actor and Critic**: *(Actor)* We searched over $\{0.01, 0.001, 0.0001, 0.0003\}$ and found that 0003 to be the most stable for the actor's learning across all tasks. *(Critic)* Similarly to actor, we searched over $\{0.01, 0.001, 0.0001, 0.0003\}$ and found that 0.0003 to be the most stable for the critic's learning across all tasks.

- **Network Size of Actor and Critic**: *(Critic)* In order for the fair comparison, we employed the same network size for the Q-network. We individually performed the architecture search on each task and found a specific network size performing the best in the task. *(Actor)* Similarly to critic, we employed the same network size for the actor components in the baseline and the cascading actors in SAVO. And, likewise, we performed the individual architecture search on each task and found a specific network size performing the best in the task.

| Hyperparameter | Mine World | MuJoCo/Adroit | RecSim |
|---|---|---|---|
| **Environment** | | | |
| Total Timesteps | 10M | 3M | 10M |
| Number of epochs | 5K | 8K | 10K |
| # Envs in Parallel | 20 | 10 | 16 |
| Episode Horizon | 100 | 1000 | 20 |
| Number of Actions | 104 | N/A | 10000 |
| True Action Dim | 4 | 5 | 30 |
| Extra Action Dim | 5 | N/A | 15 |
| **RL Training** | | | |
| Batch size | 256 | 256 | 256 |
| Buffer size | 500K | 500K | 1M |
| Actor: LR | 0.0003 | 0.0003 | 0.0003 |
| Actor: $\epsilon_{\text{start}}$ | 1 | 1 | 1 |
| Actor: $\epsilon_{\text{end}}$ | 0.01 | 0.01 | 0.01 |
| Actor: $\epsilon$ decay steps | 5M | 500K | 10M |
| Actor: $\epsilon$ in Eval | 0 | 0 | 0 |
| Actor: MLP Layers | 128_64_64_32 | 256_256 | 64_32_32_16 |
| Critic: LR | 0.0003 | 0.0003 | 0.0003 |
| Critic: $\gamma$ | 0.99 | 0.99 | 0.99 |
| Critic: $\epsilon_{\text{start}}$ | 1 | 1 | 1 |
| Critic: $\epsilon_{\text{end}}$ | 0.01 | 0.01 | 0.01 |
| Critic: $\epsilon$ decay steps | 500K | 500K | 2M |
| Critic: $\epsilon$ in Eval | 0 | 0 | 0 |
| Critic: MLP Layers | 128_128 | 256_256 | 64_32 |
| # updates per epoch | 20 | 50 | 20 |
| List Length | 3 | 3 | 3 |
| Type of List Encoder | DeepSet | DeepSet | DeepSet |
| List Encoder LR | 0.0003 | 0.0003 | 0.0003 |

Table 3: Environment/Policy-specific Hyperparameters

# I  ANALYSIS ON THE NUMBER OF ACTORS NEEDED IN SAVO

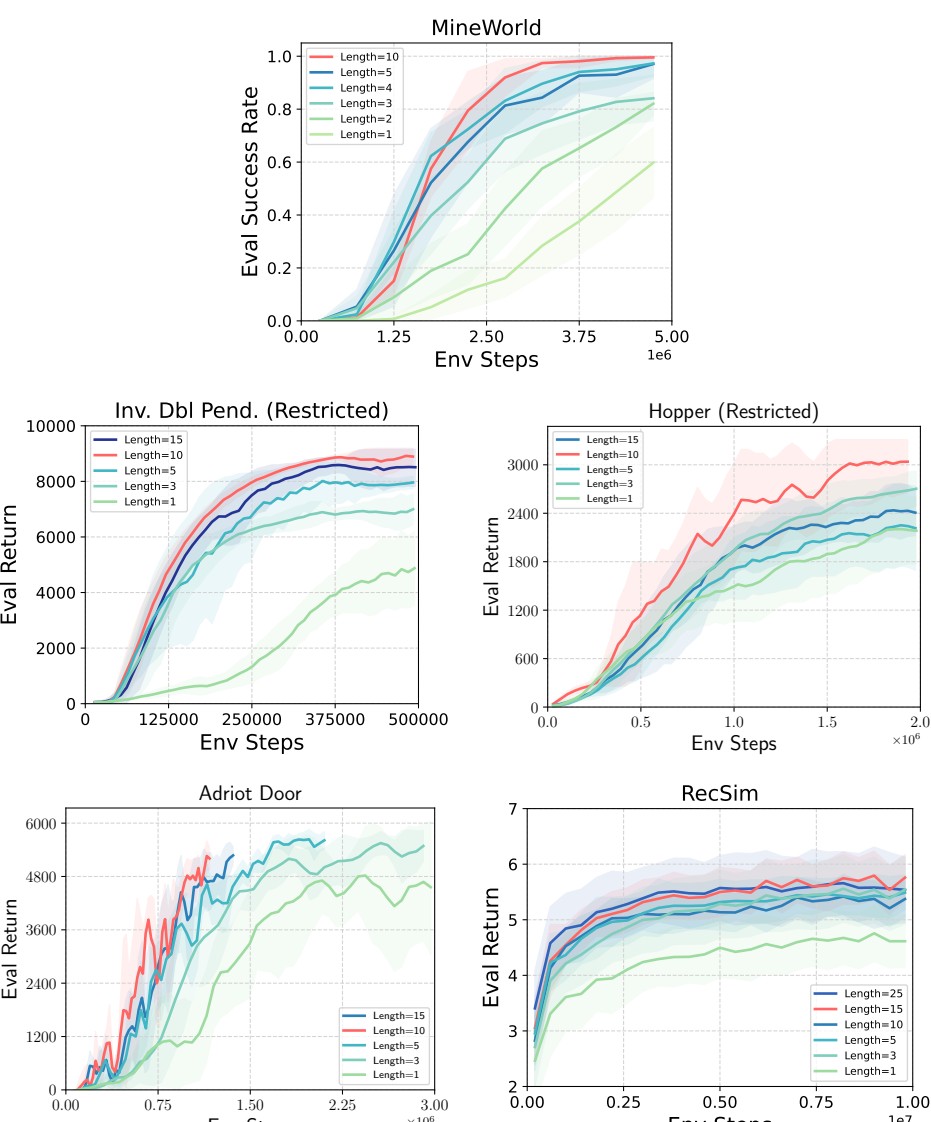

Figure 26: **Finding the optimal number of actors in SAVO**. SAVO's improvement scales well when additional actor-surrogates are added until its performance saturates and completely mitigates the suboptimality of TD3. While the gains are diminishing beyond 3-5 actors in the environments we considered, 10 actors are mostly enough to produce optimal performance (shown in red). For RecSim, which is an especially non-convex Q-landscape because of 10,000 actions and a 45-D action representation space, we note that increasing to 15 actors achieves the optimal performance.

We analyze the optimal number of necessary actors in various environments in Figure 26. As shown in Figure 11a and Figure 11b, SAVO performance improves with additional actor-surrogates until it saturates because the suboptimality of TD3 is sufficiently mitigated. In this section, we aim to propose a heuristic measure of the number of actors to use in SAVO for a new environment.

Note that all other results in the paper use a uniform actor sequence length of 3 across all environments, balancing computational cost and addressing most suboptimality, as shown in Table 1.

| Environment | Number of Actors |
|---|---|
| MineWorld | 10 |
| Inv Double Pen | 10 |
| Restricted Hopper | 10 |
| Adroit Door | 10 |
| RecSim | 15 |

Table 4: **Optimal number of actors in SAVO** to completely mitigate the suboptimality of TD3, thereby saturating performance gain, for various environments based on the analysis in Figure 26.

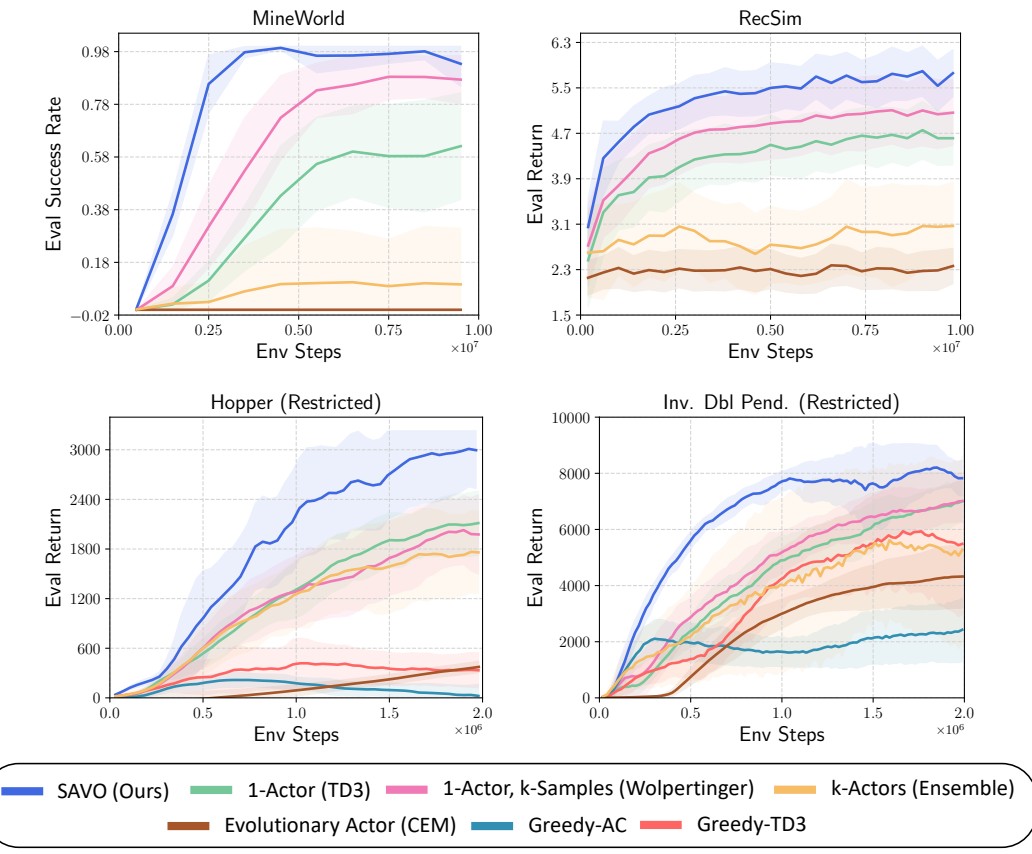

Figure 27: **SAVO optimized for number of actors against baselines**. Comparison with baselines with SAVO optimized for the hyperparameter of the number of actors (10-15 actors) shows a more significant improvement than using only 3 actors in Figure 8.

## I.1 Insights about optimal number of actors

The complexity of the underlying Q-function significantly impacts the number of additional actor-surrogate pairs required for near-optimal performance. We report the optimal number of actors beyond which performance gains saturate in Table 4 and draw the following insights:

1. **Non-convexity of the Q-landscape:** The number of necessary actors is largely influenced by how non-convex the Q-landscape is:

   - **Discrete actions:** Higher dimensionality and larger action spaces create more challenging Q-landscapes. For instance, RecSim (45-D actions, 10,000 actions) needs 15 actors, while MineWorld (9-D actions, 50 actions) requires 10 actors.

   - **Continuous actions:** Non-convexity of the Q-landscape, rather than action-space dimensionality, plays a bigger role. Restricted Hopper (3-D actions, persistent non-

convexity in all states) requires the same number of actors (10) as Adroit Door (28-D actions, non-convexity in only certain states where precise manipulation is required).

2. **Visualization of Q-landscapes:** One way to estimate the non-convexity of any environment is to study its Q-landscape visualization, such as Figure 29 and Figure 30 in Inverted Pendulum (1D action space), and Figure 31 and Figure 32 in Hopper (3D action space projected down to 2D). Such visualizations are feasible for these lower-dimensional environments but do not work for higher dimensions where 2D projections fail to preserve essential information about non-convexity.

### I.2 HEURISTIC RECOMMENDATIONS FOR ACTOR SELECTION

As a practical guideline on a new environment, we recommend using as many additional actors as the compute budget allows, so one does not need to determine the "optimal number of actors" to begin with. This works because SAVO's improvements are fairly monotonic with the number of actors. In our experiments, we observed that even 3 actors gave meaningful improvements. Below, we propose a heuristic to answer *what is the optimal number of actors* to enable SAVO to reach near-global-optimal performance.

Determining the optimal number of actors for a task is inherently tied to the non-convexity of the Q-landscape, which depends on the underlying MDP. One can follow the following steps to search for this number:

1. **Start from a benchmark:** Map the task to one of the environments analyzed in Table 4 and Figure 26. For instance, new dexterous manipulation tasks would be similar to Adroit Door and use the corresponding number of actors (i.e., 10).

2. **Apply exponential search:** Start with a small number of actors and double iteratively until performance saturates (e.g., $5 \to 10 \to 20 \to \ldots$), following exponential search.

### I.3 BENCHMARKING SAVO WITH OPTIMAL NUMBER OF ACTORS

While the results in main paper in Figure 8 use only 3 actors, we show in Figure 27 that SAVO's improvement over TD3 and other baselines is even more significant when the number of actors is optimized and chosen as 10 (or 15 in RecSim).

## J SOFT ACTOR-CRITIC (SAC): MITIGATING SUBOPTIMALITY WITH SAVO

We show that SAC (Haarnoja et al., 2018) is susceptible to gradient-descent-based local optima in the soft Q-landscape and demonstrate how SAVO improves performance when integrated with SAC.

**SAC is susceptible to local optima in soft Q-landscape.** DPG-based methods like TD3 optimize deterministic policies using:

$$\pi^* = \arg\max_\pi \mathbb{E}_{s \sim \rho^\pi} \left[ Q^\pi(s, \pi(s)) \right],$$

where gradient ascent on $Q^\pi(s, \pi(s))$ often results in convergence to local optima due to the non-convexity of the Q-landscape.

SAC extends this framework by optimizing stochastic policies through entropy regularization, as:

$$\pi^* = \arg\max_\pi \mathbb{E}_{s \sim \rho^\pi, a \sim \pi} \left[ Q^\pi(s, a) + \alpha \mathcal{H}(\pi(\cdot|s)) \right],$$

where $\mathcal{H}(\pi(\cdot|s)) = -\mathbb{E}_{a \sim \pi}[\log \pi(a|s)]$ is the entropy of the policy, weighted by $\alpha > 0$.

However, despite the entropy-regularized objective, SAC's actor is trained with gradient ascent on the soft Q-function $Q^\pi(s, a)$, which can be non-convex. Local optima in the (soft) Q-landscape arise from fundamental properties of the MDP and the non-convex relationship of actions and expected environment return. As a result, SAC policies are as prone to being trapped in local optima, in the KL-divergence sense, defined by the soft Q-landscape.

**SAVO to mitigate SAC suboptimality.** To address this challenge of SAC's stochastic actor getting stuck in the soft Q-landscape's local optima, we propose using SAVO as the actor architecture for

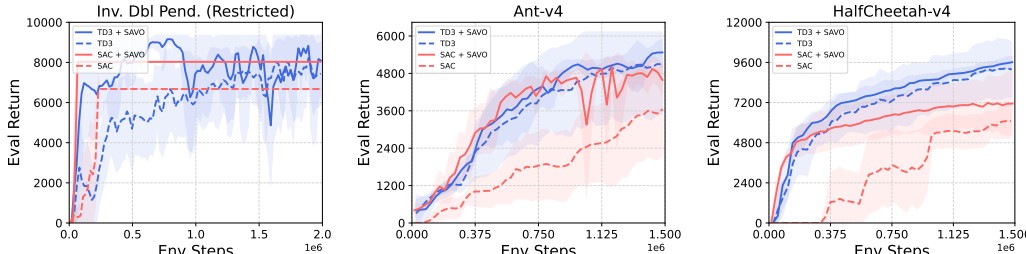

Figure 28: **SAVO is complementary to TD3 and SAC**. SAVO + SAC outperforms SAC in the three tasks evaluated: (i) Restricted Inverted Double Pendulum, (ii) Unrestricted Ant-v4, (iii) Unrestricted HalfCheetah-v4. SAVO improves or matches the performance of TD3 in the severely non-convex Q-landscape of the Restricted Inverted Double Pendulum and the high-dimensional action spaces of Ant-v4 and HalfCheetah-v4.

SAC. In our approach, we introduce a maximizer stochastic actor $\pi_M$ that selects from successive stochastic actors $\nu_i(s; a_{<i})$ by maximizing:

$$\pi_M(s) := \arg \max_{\nu_0,\ldots,\nu_k} \mathbb{E}_{s\sim\rho^\pi, a\sim\pi} \left[ Q^\pi(s, a) + \alpha \mathcal{H}(\pi(\cdot|s)) \right].$$

This SAC+SAVO approach leverages SAVO's capacity to dynamically select policies that better navigate the soft Q-landscape while preserving SAC's entropy-regularized exploration.

For this preliminary combination of SAC with SAVO, we do not employ the successive surrogates but only utilize successive actors with conditioning on previous actions.

**Empirical Results.** Figure 28 illustrates the relative performance of SAC, TD3, TD3+SAVO, and SAC+SAVO across the three tasks. Key findings include:

- *Hopper* and *Walker2D:* SAC+SAVO significantly improves performance compared to SAC, demonstrating SAVO's ability to overcome local optima in the soft Q-landscape.

- *Inverted Pendulum:* SAC+SAVO exhibits faster convergence compared to SAC, further highlighting the synergy between SAVO and entropy-regularized stochastic policies.

- Across all tasks, TD3+SAVO consistently outperforms TD3, confirming SAVO's generalizability to deterministic policy optimization.

These results underscore the effectiveness of combining SAVO with both SAC and TD3, providing a robust solution to mitigate local optima and enhance exploration in complex control tasks.

# K  Q-VALUE LANDSCAPE VISUALIZATIONS

## K.1  1-DIMENSIONAL ACTION SPACE ENVIRONMENTS

We conducted a Q-space analysis across Mujoco environments to show that successive critics reduce local optima, aiding actors in optimizing actions. The outcomes are depicted in Figures 29 and 30.

Figure 29 illustrates a representative Q landscape from the easy environments, which are uniformly smooth. This uniformity in the primary Q space simplifies the identification of optimal actions.

Figure 30 shows that the primary Q landscape (leftmost and rightmost) in challenging environments is clearly uneven with several local optima. However, the Q landscapes learned by successive critics $Q_i$ demonstrate a gradual transition toward smoothness by pruning out the locally optimal peaks below the previously selected actions' Q-values. This aids the actors in identifying improved actions that are better global optima over the primary critic. Finally, when visualized together on the primary critic (rightmost figure) the subsequent actions yield more enhanced Q-values than $a_0$, which would have been the action selected by a single actor. This translates into better evaluation performance as shown in Figure 11c and better sample efficiency as shown in Figure 8.

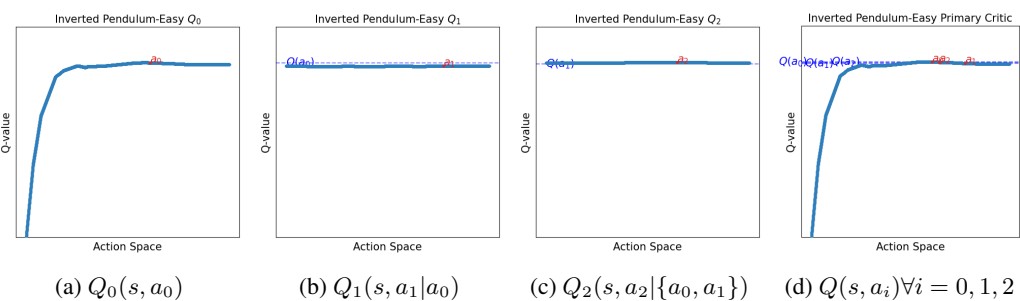

(a) $Q_0(s, a_0)$    (b) $Q_1(s, a_1|a_0)$    (c) $Q_2(s, a_2|\{a_0, a_1\})$    (d) $Q(s, a_i)\forall i = 0, 1, 2$

Figure 29: Successive Q landscape and primary Q landscape of Inverted Pendulum-v4.

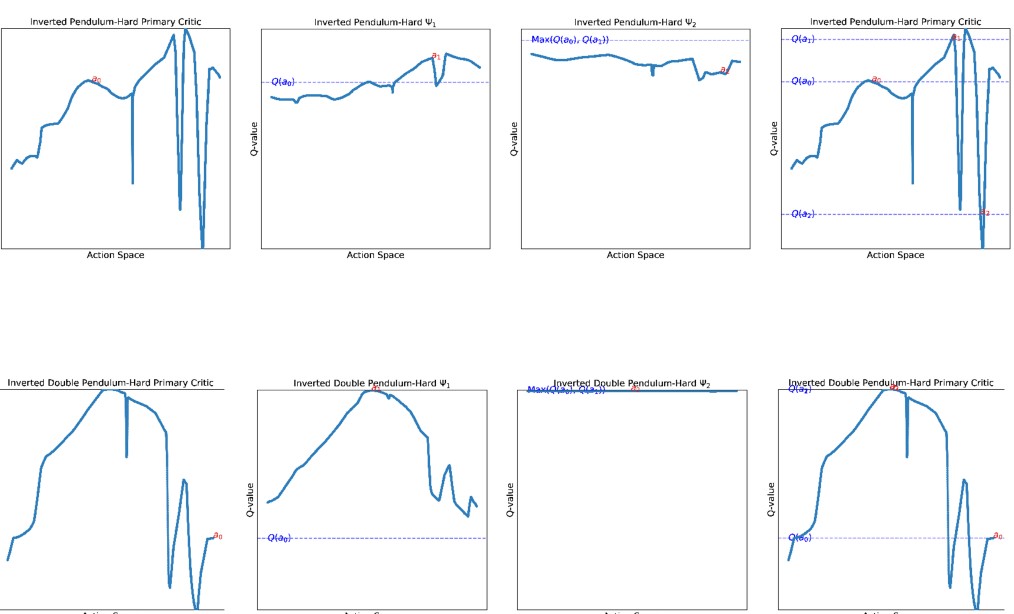

Figure 30: Successive Q landscape and primary Q landscape across different Restricted Environments.

### K.2 HIGH-DIMENSIONAL ACTION SPACE ENVIRONMENTS: HOPPER-V4

In Fig.31 and Fig. 32, we visualize Q-landscapes for a TD3 agent across different environments, starting with Hopper-v4. Here, actions from the 3D action space are projected onto a 2D plane using UMAP, with 10,000 actions sampled at equal intervals to ensure adequate coverage. These Q-values are plotted using trisurf, introducing some artificial ruggedness but providing more reliable visualizations than grid-surface plotting. Despite the inherent limitations of dimensionality reduction—where the loss of one dimension distorts distances and relative positions—the Q-landscape for Hopper-v4 reveals a large globally optimal region (in yellow), offering a clear gradient path that minimizes the risk of the gradient-based actor getting stuck in local optima.

In Hopper-Restricted, the Q-landscapes become more complex due to the restriction of actions within a hypersphere, with suboptimal peaks where gradient-based actors can potentially get trapped. Although dimensional reduction limits conclusive analysis, these landscapes appear to have more local optima compared to Hopper-v4. For higher-dimensional environments like Walker2D-v4 (6D) and Ant-v4 (8D), projecting to 2D leads to significant information loss, making it difficult to assess convexity. Despite this, Walker2D-v4 shows a large optimal region where consecutive actions produce similar outcomes, indicating that contact-based tasks like Walker2D and Hopper do not inherently induce numerous local optima. However, for more complex environments like Ant-v4 and Walker2D-Restricted, the visualizations provide limited insights due to the challenges of dimensionality reduction.

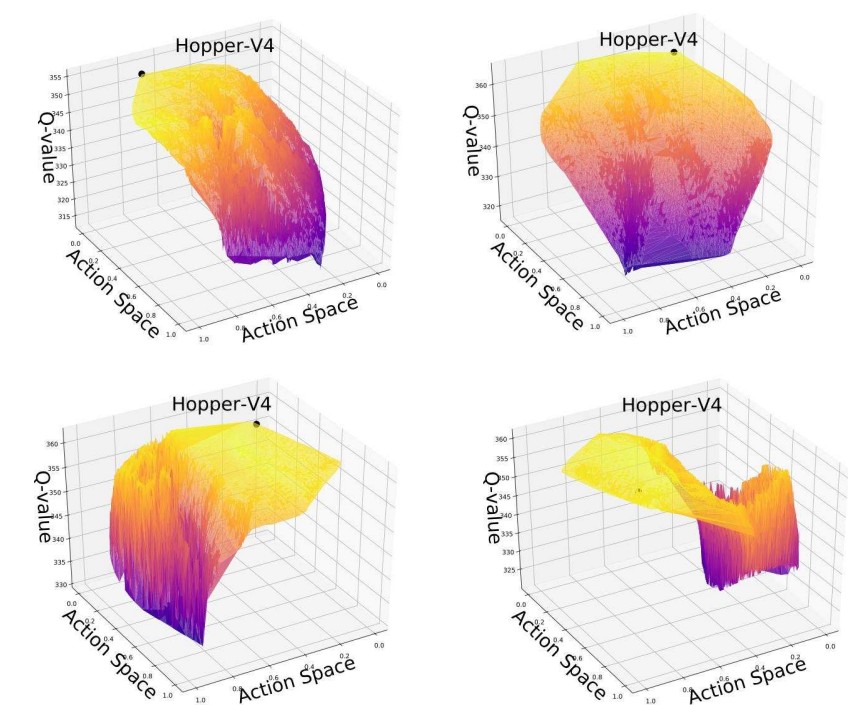

Figure 31: Hopper-v4: Q landscape visualizations at different states show a path to optimum.

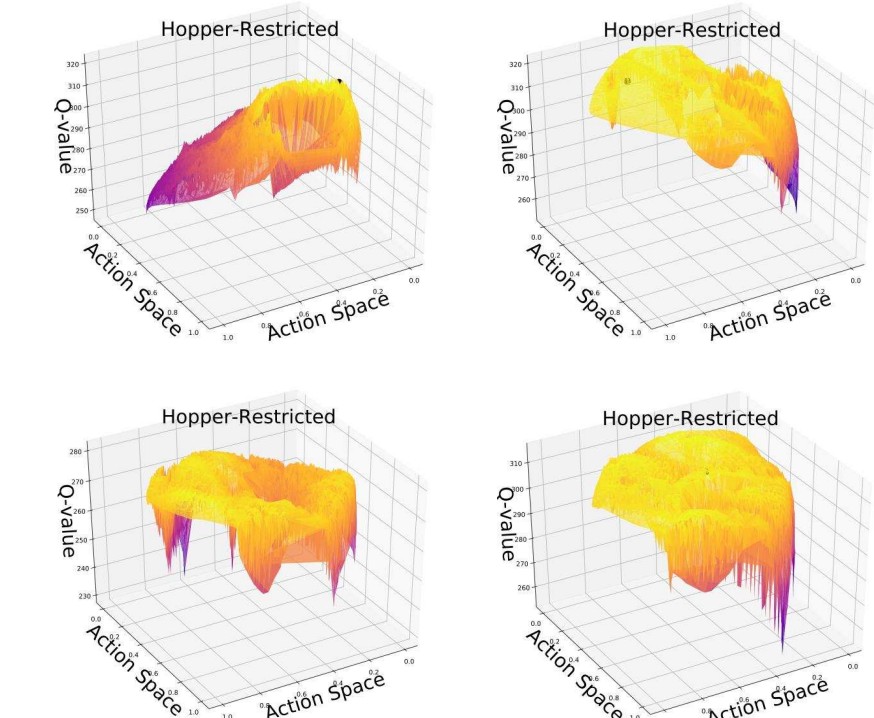

Figure 32: Hopper-restricted: Q landscape visualizations at different states show several local optima.

