# OpenReview forum: "Mitigating Suboptimality of Deterministic Policy Gradients in Complex Q-functions"
_ICLR.cc/2025/Conference — Submitted to ICLR 2025_

### Official Review · Reviewer_6956 · 2024-10-31

**Soundness:** 2
**Presentation:** 3
**Contribution:** 3
**Rating:** 8
**Confidence:** 3

**Summary:**

The paper presents a new actor architecture that combines multiple actors and simplified Q-function surrogates. The new method eases the challenge of maximizing Q-function in complex tasks like dexterous manipulation and restricted locomotion.

**Strengths:**

1. The paper has a clear motivation for the challenge of maximization Q-function in commonly used deterministic policy gradient algorithms. Figures 1-3 are all strong evidence of this problem.
2. The paper is well-written. Especially for the transition from Section 3 to Section 4, the connection is quite natural. The introduction of the main method of SAVO is also clear.
3. The extensive experiments (5 environments) also strongly prove the effectiveness of the SAVE architecture.
4. It is very appreciated that the authors also mention about the potential limitations in the inference time of the proposed method. I think it's a nice trade-off between better optima and computational efficiency.

**Weaknesses:**

1. The surrogate value function also requires a smooth version to improve performance. Is it also possible to add smoothness to the origianl Q-value? It would be helpful to add it as a baseline.

**Questions:**

1. The diversity of policy function $\nu$ and surrogate value function $\Psi$ will have a large influence on the algorithm's final performance. For the policy, do special initialization techniques increase the diversity of the policy and improve its performance? For the value function introduced in 4.4, is $Q(s, a)$ different from $Q^{\mu_M}(s,a)$ with separate neural networks? Besides, both policy and value function requires conditional input on the previous action. How to deal with the variational number of inputs?
2. If the action dimension is large, maximizing Q-function could be even harder. Do you recommend using the SAVO method could potentially require more agents?

---

> ### Author Response · Authors · 2024-11-26
> **Added Q-smoothing baseline and Specialized initialization techniques**
>
> We appreciate the reviewer for their thoughtful evaluation and valuable ideas to improve our method. We are pleased that you appreciated the motivation and evidence of the suboptimality problem, paper writing, extensive experiments, and trade-off of compute and performance. We incorporate all your suggestions below:
>
>
> &nbsp;
>
> ----
>
> ### 1. Added Q-smoothing as a baseline
>
> > The surrogate value function also requires a smooth version to improve performance. Is it also possible to add smoothness to the original Q-value? It would be helpful to add it as a baseline.
>
> Thank you for your insightful suggestion! Notably, we had already done this study and our baselines already incorporate smoothness over the Q-value in discrete action space tasks, where it was deemed necessary. Based on your question, we have added an exhaustive analysis in Appendix G.6, Figure 24, and Table 2.
>
> We empirically analyze the effect of Q-smoothing, where an auxiliary Q-function smooths the primary Q-function to ease gradient flow, just like an approximate surrogate does for the hard-threshold surrogate (Section 4.4).
>
> &nbsp;
>
>
> | Baseline                 | MineWorld | RecSim | RecSim-Data | Hopper  | Walker2D |
> |--------------------------|:---------:|:------:|:-----------:|:-------:|:--------:|
> | TD3                      | 0.6       | 4.5    | 42          | 2100    | 2700     |
> | Wolpertinger **[Naive]**     | **0.0**   | **3.9**| **40**      | 1650    | 2900     |
> | Wolpertinger **[Smooth]**    | **0.9**     | **5.0**  | **46**        | 1850    | 2400     |
> | SAVO (Ours)              | 0.98      | 5.5    | 51          | 2500    | 3200     |
>
> &nbsp;
>
>
> This table compares the performance of baselines with and without Q-smoothing across tasks. 1-Actor k-Samples or Wolpertinger [Naive] significantly underperforms in discrete action space tasks, and thus, we already reported results on Wolpertinger [Q-Smoothing]. In continuous action space tasks, there was no benefit to Q-smoothing, and thus we chose to report results on Wolpertinger [Naive] as it is closer to the underlying TD3 algorithm. Note that the **same Q-smoothing principle is already applied for TD3, SAVO, and all baselines**, i.e., their Q-function is smoothed in discrete action spaces, but unsmoothed in continuous action space tasks.
>
> 1. **Discrete Action Space results have Q-smoothing:** Q-smoothing is essential for discrete tasks, facilitating gradient flow on points in the space that lie between the action representations of true actions. This helps because an actor can query for gradients anywhere, but it is not necessary that the Q-landscape is meaningful there. Smoothing ensures gradients towards valid actions’ Q-values.
> 2. **Continuous Action Space results do not have Q-smoothing:** Q-smoothing provides no benefit in continuous tasks, as gradients are naturally preserved.
>
> &nbsp;
>
> ----
>
>
> ### 2A. Added comparison to specialized initialization for diverse actor-surrogates
>
> > The diversity of policy function and surrogate value function will have a large influence on the algorithm's final performance. For the policy, do special initialization techniques increase the diversity of the policy and improve its performance?
>
> This is an insightful suggestion! We did not explicitly enforce diversity in the actors and surrogates, and initialized the networks with standard Xavier initialization. However, based on your suggestion, we tried 2 specialized initialization strategies:
> - **Random**: weights are initialized from standard Normal distribution, instead of Xavier. ⇒ w = noise $\sim \mathcal{N}(0, 1)$
> - **Add**: weights are initialized with Xavier, but then standard normal noise is added. ⇒ w = Xavier-init + noise $\sim 0.5 * \mathcal{N}(0, 1)$
>
> &nbsp;
>
>
> The reward curves are reported in Figure 25 and their performance summarized below:
> 1. **MineWorld**: Add $\approx$ Random $\approx$ Xavier
> 2. **RecSim**: Add $\approx$ Random $\approx$ Xavier
> 3. **Hopper (Restricted)**: Add $\approx$ Random $\approx$ Xavier
> 4. **Adroit Door** : Add $\approx$ Random < **Xavier**
>
> Thus, **specialized initialization to enforce larger diversity did not boost performance**. However, we believe incorporating *diversity as an explicit objective could be a helpful heuristic* and will investigate it in the future, thank you again for a great suggestion!

---

> > ### Author Response · Authors · 2024-11-26
> > **Added analysis on the optimal number of agents in SAVO**
> >
> > ### 2B. Variable number of inputs are handled using list summarizers like DeepSet
> >
> > > For the value function introduced in 4.4, is $Q(s, a)$ different from $Q^{\mu_M}(s, a)$ with separate neural networks?
> >
> > No, they are the same. There is only one true Q-function $Q = Q^{\mu_M}$ function, i.e., a neural network that estimates the expected return of our agent $\mu_M$. We explicitly included the superscript here so as to distinguish it from $Q^\mu(s, a)$, which is what the Q-function estimates in the base algorithm TD3, without SAVO.
> >
> > &nbsp;
> >
> > > Besides, both policy and value function requires conditional input on the previous action. How to deal with the variational number of inputs?
> >
> > The conditioning on variable number of previous actions is done in two steps (SAVO architecture in Figure 4 shows these components visually):
> >
> > **1. List-summarizer**
> > - **Approach**: Summarize the preceding action list (with state conditioning) using a list-compatible architecture like a DeepSet (or bi-lstm or transformer), described in Appendix E.3: $\phi_{\text{summary}} = \lbrace (a_0, s), (a_1, s), ..., (a_{i-1}, s) \rbrace$.
> > - Note: We concatenate each action with state like $(a_0, s)$ before summarizing with DeepSet, so that the summary latent can capture the effect of previous actions with **current state as the context**.
> > - **Experiment**: Figure 20 shows that SAVO is robust to the choice of list summarizer: *DeepSet, LSTM, Transformer — all work well*.
> >
> > **2. FiLM conditioning**
> > - **Approach**: condition the actor $\nu_k(s; a_{<i})$ and surrogate network $\Psi_k(s, a; a_{<i})$ on $\phi_{\text{summary}}$ via FiLM modulation, described in Appendix E.4: $\nu_k(s; \phi_{\text{summary}})$ and $\Psi_k(s, a; \phi_{\text{summary}})$
> > - **Experiment**: Figure 21 shows that *FiLM conditioning is more effective* than just concatenating the summary with the state at input.
> >
> >
> > &nbsp;
> >
> > ----
> >
> > &nbsp;
> >
> > ### 3. Added analysis on the optimal number of agents in SAVO
> > > If the action dimension is large, maximizing Q-function could be even harder. Do you recommend using the SAVO method could potentially require more agents?
> >
> > Yes, SAVO can benefit from more agents, and we have added an exhaustive study based on your and Reviewer S3Pa’s recommendation. We analyze how many actors are needed to reach near-optimal performance in the various tasks we have in Figure 26 by running SAVO with various numbers of actors. In the below table, we list the optimal number of actors beyond which the performance gain saturates (original results are on 3 actors).
> >
> > | Environment | Number of actors |
> > |:---:|:---:|
> > | MineWorld | 10 |
> > | Inv Double Pen | 10 |
> > | Restricted Hopper | 10 |
> > | Adroit Door | 10 |
> > | RecSim | 15 |
> >
> > &nbsp;
> >
> >
> > **[Key insights]**
> > - The number of actors needed is a **function of how severely non-convex** the underlying Q-landscape is (not necessarily the action-space dimensionality).
> > - **For discrete actions, a larger action space and a larger dimensionality of action representations** would likely imply a more challenging Q-landscape. Thus, RecSim with 10,000 actions and 45-D action representations needs 15 actors while MineWorld with 50 actions and 9-D action representations needs only 4 actors.
> > - **For continuous actions**, it is less a matter of how high-dimensional the action space is, but **how non-convex is the Q-landscape on different states** of the environment. Therefore, Restricted Hopper has a 3-D action space but has non-convexity in every state (because of restrictions) requires the equal number of actors as Adroit Door which has a 28-D action space but has non-convexity only in states that require precise manipulation.
> >
> > &nbsp;
> >
> >
> > **[Heuristic Recommendation]**
> >
> > Determining the optimal number of actors for a task is inherently tied to the non-convexity of the Q-landscape, which depends on the underlying MDP. As a practical guideline, we recommend using 5 or 10 actors depending on the balance of available compute time and desired improvement.
> >
> > &nbsp;
> >
> >
> > **[Recipe for finding the optimal number of actors]**
> >
> > 1. Start from a benchmark: Map the task to one of the environments analyzed in Table 4 and Figure 26. For instance, new dexterous manipulation tasks would be similar to Adroit Door and use the corresponding number of actors (i.e., 10).
> > 2. Apply exponential search: Start with a small number of actors and double iteratively until performance saturates (e.g., 5 → 10 → 20 → . . . )
> >
> >
> > ----
> >
> > &nbsp;
> > &nbsp;
> >
> >
> > We thank you for your promising ideas on improving and extending SAVO. We believe your suggestions on Q-smoothing, diversifying actor-surrogates, and scaling to more actors are all promising avenues to improve SAVO further, and incorporating these analyses in the paper has improved the strength and future promise of our work.
> >
> > We hope we've addressed your comments adequately with extensive experimentation. We are happy to answer any further questions and incorporate any further suggestions.

---

> > > ### Comment · Reviewer_6956 · 2024-12-02
> > > **Response to rebuttals**
> > >
> > > Thank the authors for their clear responses and added experiments, which answered my previous questions and helped me further understand the effectiveness of the methods, especially the smoothness of the Q-function and the number of agents. Overall, this paper is motivated and well-supported by all experiments. I will raise my score to 8.

---

### Official Review · Reviewer_S3Pa · 2024-10-31

**Soundness:** 2
**Presentation:** 2
**Contribution:** 3
**Rating:** 5
**Confidence:** 4

**Summary:**

Summary Of The Paper:

The paper develops a method to mitigate actors from getting stuck in a locally optimal Q value. Their proposed method considers producing surrogate estimates of the Q landscape based on the last seen maximum Q value, and restricts the actor from making actions in local optima through a successive actor architecture.

**Strengths:**

The paper is easy to read, and generally well written and is well motivated by the issue of converging to a local optima.

The idea of restricting local optimal using tabu search in this successive manner is very interesting.

The idea appears fairly novel. The closest work to this is probably SAC, although this paper is quite a bit different in motivation. This helps with novelty.

**Weaknesses:**

### Weaknesses

1. The complexity of the true underlying Q function should affect how many surrogates are needed. There should be an analysis or heuristic on how to determine the optimal or necessary number of surrogates needed for a particular task. This will also determine the computational viability of SAVO, in situations where we might require several surrogates to converge to global optima.

2. The experiment environments are quite simplistic with only one set of 3D environments. There needs to be more analysis on how this method works with high dimensional continuous state and action spaces, especially because here Q would be a function of these high dimensional parameters. In SAC’s original paper, they baseline with Ant and humanoid, and SAVO should be compared to those environments as well. Since SAC is another method that proposes to exit local optima with the stochastic actor and entropy objective, a full comparison between SAV + DDPG/TD3 and SAC (+ SAVO) should be done.

3. Surrogates are trained via L2 loss. How do we know if the surrogates have approximated modified Q function sufficiently? What if there are margins of errors that potentially result in the surrogates converging to local optima. Then how do we diagnose or detect these problems? It seems like SAVO involves increasing the number of hyperparameters that require tuning. How sensitive is the training regime of the surrogates, and at what point do they yield no benefit compared to the baseline architectures? How do the design choices of the surrogates impact the stability of training? This method adds a lot of overhead on top of something like DDPG. That's my biggest worry about the paper.

4. It would be nice to better understand the generalizability of SAVO to other environmental regimes. How does SAVO perform in sparse reward environments? It seems that SAVO is highly dependent on how well we estimate the truncated Q through surrogates. It would be helpful to see how SAVO can help with improving HER or other actor critic architectures.

**Questions:**

Section 4.3 defines \nabla \Psi as a function of \nabla Q. Is \Psi simply Q + constant with the truncation term?

---

> ### Author Response · Authors · 2024-11-26
> **1. Added analysis on the optimal number of surrogates**
>
> We sincerely thank the reviewer for their thoughtful evaluation and valuable feedback. We are pleased that you appreciated the paper writing, motivation of the suboptimality problem, the tabu-inspired approach, and overall novelty. We address your concerns below.
>
>
> &nbsp;
>
> ----
>
> ### 1. Added analysis on the optimal number of surrogates
>
> &nbsp;
>
>
> > 1. The complexity of the true underlying Q function should affect how many surrogates are needed. There should be an analysis or heuristic on how to determine the optimal or necessary number of surrogates needed for a particular task. This will also determine the computational viability of SAVO, in situations where we might require several surrogates to converge to global optima.
>
> Thank you for this great suggestion! We have run an exhaustive study to analyze how many surrogates are needed to reach near-optimal performance in the various tasks we have. Appendix Figure 26 plots the performance by ablating the length of the actor sequence on various environments. In the below table, we list the optimal number of actors beyond which the performance gain saturates. Note that the number of surrogates = length of actor sequence - 1.
>
> | Environment | Number of actors |
> |:---:|:---:|
> | MineWorld | 10 |
> | Inv Double Pen | 10 |
> | Restricted Hopper | 10 |
> | Adroit Door | 10 |
> | RecSim | 15 |
>
> Note: all other results in the paper were reported on a length of 3, across all environments, to keep a uniform hyperparameter across all tasks that addresses most of the suboptimality with a small amount of extra compute (Table 1).
>
> **[Key insights]**
> - The number of actors needed is a **function of how severely non-convex** the underlying Q-landscape is.
> - For discrete actions, a **larger action space** and a larger dimensionality of action representations would likely imply a more challenging Q-landscape. Thus, RecSim with 10,000 actions and 45-D action representations needs 15 actors while MineWorld with 50 actions and 9-D action representations needs only 4 actors.
> - For continuous actions, it is less a matter of how high-dimensional the action space is, but how **non-convex is the Q-landscape on different states** of the environment. Therefore, Restricted Hopper has a 3-D action space but has non-convexity on every state (because of restrictions) requires the equal number of actors as Adroit Door which has a 28-D action space but has non-convexity only in states that require precise manipulation.
>
> **[Heuristic Recommendation]**
> We agree with the reviewer that it would be valuable to pre-determine the number of actors needed, but pre-determining this hyperparameter would be an open challenge. This number scales with the non-convexity of the underlying Q-function across all states of the environment, which is a property of the underlying MDP. Unfortunately, it is hard to make an accurate estimation of this non-convexity, because environments can require different kinds of decision-making at different states and the Q-landscape could look very different even within a single environment. This applies to any method trying to optimize these Q-landscapes. For instance, CEM would require more iterations of refinement of the action distribution and SAVO would require more actors.
>
> We attempt to understand the non-convexity of Q-landscapes via visualizations shown in Figures 29-32 in 1-D action space and 3-D action space (by projecting it to 2D). But, such visualization is not possible in higher dimensional environments, because the 2D projection cannot preserve the information about non-convexity.
>
> **[Recipe for finding the optimal number of actors]**
> 1. For any given task, we recommend mapping it to the closest environment that we have analyzed, such as Adroit Door for a manipulation task. Then, we start with that number of actors, such as 5 or 10.
> 2. Next, we can employ exponential search or doubling search, where the number of actors is doubled until diminishing returns: 5 → 10 → 20 → ...

---

> ### Author Response · Authors · 2024-11-26
> **2. Added: SAC + SAVO outperforms SAC**
>
> ### 2. Added experiment: SAC+SAVO > SAC
>
> > The experiment environments are quite simplistic with only one set of 3D environments. There needs to be more analysis on how this method works with high dimensional continuous state and action spaces, especially because here Q would be a function of these high dimensional parameters. In SAC’s original paper, they baseline with Ant and humanoid, and SAVO should be compared to those environments as well.
>
> We highlight the existing and new results to address this concern:
> - The 3 Adroit environments have a high-dimensional **28-D action space** and RecSim and RecSim-data have **45-D** action representation space.
> - Appendix reports results on **unrestricted locomotion in Ant-v4 and HalfCheetah-v4 in Figure 15** (we did not create restricted versions of these environments). SAVO matches the performance of TD3 here, but does not improve, likely because there is no suboptimality to mitigate — for unrestricted Inverted Pendulum and Hopper, the Q-landscapes are fairly convex (see Figures 29, 31).
> - In the new results, SAC underperforms in Ant and HalfCheetah, and **SAC+SAVO mitigates SAC's suboptimality** successfully in Figure 28 (details below).
>
> &nbsp;
>
> > Since SAC is another method that proposes to exit local optima with the stochastic actor and entropy objective, a full comparison between SAVO + DDPG/TD3 and SAC (+ SAVO) should be done.
>
> We appreciate your insightful comment. In response, we have extended our analysis to include both theoretical explanations and experimental results demonstrating that
> A. SAC also suffers from gradient-descent-based local optima in the soft Q-landscape, and
> B. SAC+SAVO enhances SAC's performance in high-dimensional continuous state and action spaces.
>
> **[A. SAC is susceptible to local optima in soft Q-landscape]**
>
> DPG-based methods like TD3 optimize the following maximization problem with gradient ascent:
> $$
> \pi^* = \arg \max_{\pi} \mathbb{E}_{s \sim \rho^\pi} \left[ Q^\pi(s, \pi(s)) \right],
> $$
>
> In contrast, SAC learns stochastic policies and augments the objective to encourage exploration through entropy maximization, optimizing:
> $$
> \pi^* = \arg \max_{\pi} \mathbb{E}_{s \sim \rho^\pi, a \sim \pi} \left[ Q^\pi(s, a) + \alpha \mathcal{H}(\pi(\cdot | s)) \right],
> $$
>
> where $\mathcal{H}(\pi(\cdot | s)) = -\mathbb{E}_{a \sim \pi} [\log \pi(a | s)]$ is the entropy of the policy, and $\alpha > 0$ controls the weight of the entropy term.
>
> &nbsp;
>
> $\implies$ Even with entropy regularization, SAC's policy gradients still rely on the **soft Q-function $Q^\pi(s, a)$, which can possess local optima** due to inherent non-convexity of the MDP’s reward structure. This non-convexity arises from complex relationships between the agent’s actions and the environment rewards. Consequently, SAC’s stochastic actor can get stuck in the local optima (in the KL-divergence sense) within the soft Q-landscape, limiting its performance.
>
> &nbsp;
> &nbsp;
>
> **[B. SAVO+SAC outperforms SAC]**
>
> We perform a preliminary analysis of combining SAVO with SAC, where we learn a **maximizer stochastic actor** $\pi_M$ that selects from successive stochastic actors $\nu_i(s; a_{<i})$ with the entropy regularized objective.
> $$
> \pi_M(s) := \arg \max_{\nu_0, ..., \nu_k} \mathbb{E}_{s \sim \rho^\pi, a \sim \pi} \left[ Q^\pi(s, a) + \alpha \mathcal{H}(\pi(\cdot | s)) \right],
> $$
>
> We evaluated this SAC+SAVO approach alongside SAC in three continuous control environments: **Restricted Inverted Double Pendulum, unrestricted Ant-v4, and unrestricted HalfCheetah-v4**. Figure 28 shows SAC is suboptimal (note that we directly use stable-baselines3 SAC implementation), even worse than TD3, but SAVO+SAC is significantly better. We draw these key insights:
> - *Improved Performance*: SAC+SAVO consistently outperforms SAC in all evaluated environments, indicating that SAVO effectively mitigates the susceptibility of SAC to local optima.
> - *High-Dimensional Action Spaces*: SAC successfully improves suboptimality, when present, in high-dimensional action spaces like Ant-v4 and HalfCheetah-v4.
> - *SAC improvements are orthogonal to SAVO*: Figure 18 already showed that in different restricted locomotion tasks, SAC may or may not be better than TD3, but SAVO+TD3 is always better than TD3, mitigating its suboptimality in non-convex Q-landscapes.

---

> > ### Author Response · Authors · 2024-11-26
> > **3. Added analysis on accuracy of surrogate approximation**
> >
> > ### 3a. Added complete analysis on accuracy of surrogate approximation
> >
> >
> > > Surrogates are trained via L2 loss. How do we know if the surrogates have approximated modified Q function sufficiently? What if there are margins of errors that potentially result in the surrogates converging to local optima. Then how do we diagnose or detect these problems?
> >
> > Thank you for this great suggestion (also: Reviewer PEvp)! We have added this in Appendix G.5 and plotted $\frac{\text{Surrogate Approximation Error}}{\text{Bellman Error}} \\%$ for all environments in Figure 23, where the numerator is $\mathcal{L}_\text{approx}$ from Eq. 9. Findings:
> >
> > - **Low Error:** Surrogates converge to 1—10\% of the Bellman error, successfully tracking Q-function changes.
> > - **Non-zero error ensures smoothness:** Non-zero error ensures gradients propagate in flat regions, preventing stagnation, as proposed in Section 4.4.
> > - **Specific Case:** For Inverted Double Pendulum (Restricted), the error increased towards the end. This happens because the agent had already converged at 50% of training, and the agent performance started deteriorating, making all the metrics unstable.
> >
> > This analysis confirms surrogates remain accurate, simplify Q-landscapes, and enable robust gradient flow. Empirically, we showed that this approximation is necessary because SAVO - approximation underperforms in Figures 7b, 22.
> >
> > ----
> >
> > &nbsp;
> >
> > ### 3b. Experiments justify every SAVO component and design choice
> > >  It seems like SAVO involves increasing the number of hyperparameters that require tuning. How sensitive is the training regime of the surrogates, and at what point do they yield no benefit compared to the baseline architectures? How do the design choices of the surrogates impact the stability of training? This method adds a lot of overhead on top of something like DDPG. That's my biggest worry about the paper.
> >
> > Thank you for explaining your concern in detail. To clarify, **SAVO surrogates do not introduce any hyperparameter** because surrogate training *follows the network architecture and hyperparameters of training the Q-function* itself. One could consider the number of actors in SAVO to be a hyperparameter, but we have kept it fixed to be 3, and now analyzed its influence as per your recommendation.
> >
> > Every single design choice introduced in SAVO is well-motivated and backed by ablation experiments, namely:
> >
> > - **Ablations** (Figures 7b, 22)
> >   + [*Smooth surrogate approximation*] enables gradient flow (Figure 9 validates visually)
> >   + [*Conditioning on previous actions*] helps
> >   + [*TD3’s action smoothing should be removed*] in non-convex Q-landscapes
> >   + [*Successive Architecture*] is better than a one-shot joint architecture
> >
> > - In fact, we show that **SAVO is robust to various design choices**:
> >   + [*Any action summarizer works*] among DeepSet, LSTM, Transformer (Figure 20)
> >   + [*Any exploration noise works*]: OU, Gaussian (Figure 17)
> >   + [*No special initialization*] is needed (Figure 25)
> >   + [*FiLM conditioning helps*], but is not necessary (Figure 21)
> >
> > We acknowledge that SAVO adds overhead in the form of additional actors and surrogates, but their implementation directly follows TD3’s actor and critic. Methods like TD3 also extended DDPG with many more hyperparameters, but it is justified because it addresses crucial function approximation challenges. Likewise, SAVO addresses TD3’s local optima challenge, with additional overhead in implementation but not as many new hyperparameters.
> >
> > &nbsp;
> >
> > Since SAVO does not require any new hyperparameter-tuning beyond TD3, we have been easily able to test SAVO on 14+ tasks with consistent improvements or matching performance:
> > - 3 discrete action space, with variations like 10k, 100k, 500k actions
> > - 4 restricted locomotion
> > - 4 unrestricted locomotion
> > - 3 dexterous manipulation
> >
> > ----
> >
> > Please let us know if there are any specific tuning concerns of SAVO, and we’d be happy to address it with discussion and experiments, or acknowledge in limitation.

---

> > > ### Author Response · Authors · 2024-11-26
> > > **4. SAVO is generalizable to various DPG-based RL algorithms because it only modifies the actor architecture**
> > >
> > > ### 4. Generalizability of SAVO to various settings
> > >
> > > > It would be nice to better understand the generalizability of SAVO to other environmental regimes. How does SAVO perform in sparse reward environments? It seems that SAVO is highly dependent on how well we estimate the truncated Q through surrogates. It would be helpful to see how SAVO can help with improving HER or other actor critic architectures.
> > >
> > > Thanks to the reviewers’ suggestion, our new analysis on surrogate approximation error (Appendix G.5, Figure 23) shows that the *surrogates are able to sufficiently approximate the Q-landscape in all environments*. We hope the above discussion shows why accurate estimation is not surprising because **surrogates are updated whenever the Q-function is updated** and once more when a gradient query is made. Likewise, in sparse reward environments, surrogates can track the Q-function directly.
> > >
> > > &nbsp;
> > >
> > > About other actor-critic architectures, we have included SAVO+SAC and shown how SAVO addresses the challenge of local optima in both TD3 and SAC. Since HER is an orthogonal improvement that improves goal-conditioned RL, we expect SAVO to be complementary to it. Unfortunately, within the scope of one paper, it is hard to evaluate with the vast RL literature on a variety of specific domains, but we highlight that **SAVO complements such methods conceptually, because it only modifies the actor architecture**.
> > >
> > > &nbsp;
> > >
> > > To recall **our contribution**, we:
> > > - demonstrate that Deterministic Policy Gradient is susceptible to the actor getting stuck in local optima,
> > > - proposed environments where this is visible,
> > > - visualized non-convexity in Q-landscapes,
> > > - proposed a novel solution (SAVO) with quantitative and visual evidence of mitigating local optima.
> > >
> > > We believe our experiments sufficiently justify these claims. However, we can attempt HER experiments if the reviewer believes that this is necessary to justify SAVO further.
> > >
> > > &nbsp;
> > > &nbsp;
> > >
> > >
> > > ----
> > >
> > > > Section 4.3 defines \nabla \Psi as a function of \nabla Q. Is \Psi simply Q + constant with the truncation term?
> > >
> > > Yes, $\Psi_i (s, a; a_{<i}) = \max \lbrace Q(s, a), \tau \rbrace$, where $\tau$ is the constant threshold, determined by the previous actions $a_{<i}$, i.e., $\tau = \max_{j<i} Q(s, a_j) $. Note that each $Q(s, a_j)$ is detached from its computation graph and only their final values are used to compute the value of $\tau$.
> > >
> > > ----
> > >
> > > &nbsp;
> > > &nbsp;
> > >
> > > We thank you for your valuable suggestions that helped refine our paper, and hope we've addressed your comments adequately with extensive experimentation. We are happy to answer any further questions and incorporate any further suggestions.

---

> > > > ### Comment · Reviewer_S3Pa · 2024-11-27
> > > >
> > > > Thank you for your response.
> > > >
> > > > Overall, my main struggle with this paper is the additional complexity it introduces.
> > > >
> > > > The gains over simpler methods like SAC/TD3 need stronger justification given this overhead.
> > > >
> > > > Namely, the recipe for determining optimal actor count still requires significant trial and error. I understand that perhaps a variety of values work here. But now I need to run the algorithm many different times, or at the very least monitor performance with respect to this extra externality.
> > > >
> > > > The response, while comprehensive, actually highlights this complexity rather than alleviating the concern.
> > > >
> > > > It looks like the proposed heuristic would exponentially increase in cost in trying to find the optimal number of surrogates. I'm particularly hesitant about the heuristic of finding a "closest" environment, because I don't really know what that means for example, if I wanted to apply this to a robotic arm. To make this type of heuristic should require even more extensive experimentation on a variety of environments.
> > > >
> > > > While SAC + SAVO does improve upon SAC, the bands in Figure 28 show quite some overlap. Is the extra overhead going to be worth it?
> > > >
> > > > Thus, I will elect to keep the original score in this case.

---

> > > > > ### Author Response · Authors · 2024-11-28
> > > > > **No need to determine "minimal" actor count: simply use as many as you can**
> > > > >
> > > > > Thank you for your feedback. We'd like to clarify what we believe is a misunderstanding about "additional complexity".
> > > > > &nbsp;
> > > > >
> > > > > 1. **No Need to Determine the Minimal Actor Count**
> > > > >
> > > > > It's **not necessary** to find an optimal number of actors for SAVO to be effective — more is better. The analysis we did is about the "minimal" number of actors that is sufficient, but one can always use more. We used only **two additional actors** for all 14 environments **without tuning**, and consistently improved over TD3 and SAC in non-convex tasks.
> > > > >
> > > > > The number of actors is not a sensitive hyperparameter like learning rate; as shown in Figure 26, the **performance improvement with more actors is near-monotonic**, so larger is better. While finding global optima in general non-convex functions is known to be impossible \[1\], SAVO provides a practical solution to *enhance* performance with **any number of actors that one’s computational resources allow without requiring re-running or tuning.**
> > > > >
> > > > > Even when it may be desired to find the minimal number of actors, the **“exponential search”** heuristic does not introduce an exponential cost to the time-complexity, **as it is an O(log n)** method \[2\] where n is the target optimal number. This heuristic is commonly used for unbounded searches in sorted arrays, which is true for our case, because SAVO performance is nearly a monotonically increasing function of the number of actors. Overall, we thank the reviewer for this insightful auxiliary analysis about the minimal number of actors for SAVO to reach saturated performance in various environments (found to be 10).
> > > > >
> > > > > However, the simplest practical guideline for SAVO is to use as many additional actors as the compute budget allows (like we used 2), so one does not need to determine the “optimal or minimal number of actors” to begin with (we have revised Appendix I.2).
> > > > >
> > > > > &nbsp;
> > > > >
> > > > > 2. **Compute-time overhead enables reliability, sample efficiency, and performance**
> > > > >
> > > > > We believe using SAVO is worth it in complex tasks with non-convex Q-landscapes, where we have shown that **TD3 and SAC are suboptimal and unreliable**. Some seeds of TD3 / SAC can solve the task optimally if gradient descent *happens to arrive* at a good optimum (1 seed overlaps in Figure 28). However, SAVO is *guaranteed* (see Theorem 4.1) to improve their reliability and final performance by enhancing the optimality of every action decision.
> > > > >
> > > > > Higher reliability and better performance are crucial in tasks where data collection is slow or expensive, such as robotics, autonomous driving, human-computer interaction, and medical applications. For instance, a 10\% sample-efficiency improvement would save thousands of hours of data collection, for a few extra hours of training due to the additional actors \[3\]. Also, a robot that's 85% successful vs 95% successful could make the difference between one that people will use and one that cannot be deployed.
> > > > >
> > > > > \[1\] Jain, Prateek, and Purushottam Kar., 2017\.  "Non-convex optimization for machine learning."
> > > > >
> > > > > \[2\] [https://www.geeksforgeeks.org/exponential-search/](https://www.geeksforgeeks.org/exponential-search/#)
> > > > >
> > > > > \[3\] Kiran, B. Ravi, et al., 2021\.  "Deep reinforcement learning for autonomous driving: A survey."
> > > > >
> > > > > ----
> > > > >
> > > > > &nbsp;
> > > > >
> > > > > We thank you for your valuable suggestions that we have all added: analyses on surrogate accuracy, SAC+SAVO, and near-monotonic scaling. Could you suggest how we could add anything else that would improve the justification of our paper? We would love to discuss this matter as much as possible.

---

> > > > > ### Author Response · Authors · 2024-12-02
> > > > >
> > > > > We appreciate your time and effort in refining our work. Regarding the concern about the “number of actors,” we have clarified that it **requires no additional tuning complexity or overhead**. A simple guideline is to use more actors, as performance improves near-monotonically with additional actors (Fig. 26). Notably, even SAVO with two additional actors (Figs. 7, 8) significantly outperforms the baselines.
> > > > >
> > > > > Since it is the last day of discussion, we would be grateful for any suggestions on further refining the justification of our work. Thank you.
> > > > >
> > > > > &nbsp;
> > > > >
> > > > > ----
> > > > >
> > > > > &nbsp;
> > > > >
> > > > > To summarize the key improvements made in response to your feedback:
> > > > > 1. Clarified that SAVO introduces `no additional hyperparameters`, because actor-surrogate training inherits the hyperparameters of the underlying RL algorithm (TD3 / SAC). SAVO `complements off-policy RL` like TD3 and SAC by only modifying the actor architecture without hyperparameter-tuning to enhance the policy improvement step of GPI.
> > > > > 2. `Appendix I`: Added analysis on the minimal number of actors for SAVO to mitigate TD3’s suboptimally fully, which helps study the degree of non-convexity of various tasks. Performance increases near-monotonically.
> > > > > 3. `Appendix J`: Added SAC+SAVO experiments, outperforming SAC in tasks like Ant and HalfCheetah.
> > > > > 4. `Appendix G.5`: Plotted surrogate approximation loss over training in all environments, showing that SAVO stays up-to-date to true Q-value throughout training.

---

### Official Review · Reviewer_PEvp · 2024-11-05

**Soundness:** 3
**Presentation:** 2
**Contribution:** 2
**Rating:** 5
**Confidence:** 2

**Summary:**

This paper introduces the *Successive Actors for Value Optimization (SAVO)* algorithm, addressing the non-convex Q-function landscapes. The claim is that in high-dimensional and multimodal action spaces, Q-functions often exhibit numerous local optima, making it difficult for gradient ascent-based actors to discover globally optimal actions. This issue can lead to poor sample efficiency and suboptimal policy performance, even when the Q-function is accurate.

SAVO tackles both continuous and discrete action settings (recommender systems). However discrete actions are mapped to a continuous representation space.

SAVO employs two main techniques:
- **Maximizer Actor**: It instroduces a maximizer actor $\mu_M(s)$ that selects actions from a set of action proposals based on the higher Q-value. Proof has been provied that this method converges in the tabular setting and show that $\mu_M(s)$ is at least as good as the base policy $\mu$ based on the policy improvement theorem. This leads to a multi-actor architecture.
- **Successive Surrogates to Simplify the Q-Landscape**: Inspired by tabu search, SAVO progressively filters out suboptimal regions in the Q-landscape. By “tabu-ing” low Q-value areas, it reduces the number of local optima, thereby making gradient ascent more effective.

**Strengths:**

- Ensemble of Actors: The ensemble-style actor, which combines policies and selects actions based on the highest Q-values, is novel afaik.
-  The surrogate Q-functions inspired by tabu search could potentially improve gradient-based optimization by reducing the number of local optima in the action space.
- The convergence proof for the maximizer actor in the tabular setting has been provided, although I did not have time to check in-depth.

**Weaknesses:**

- Quality of Action Proposals: I'm wondering whether the quality of action proposals from additional policies $v_i$ could affect the effectiveness of the maximizer actor? If these proposals are not close to the optimal actions, the maximizer actor’s effectiveness is limited, as it can only select from the actions provided. In other words, the performance of the maximizer actor is capped by the quality of these action proposals.

- The effect of smoothing in highly dynamical environments : In rapidly changing Q-landscapes, the surrogate-based tabu regions may become outdated, potentially causing the actor to miss newly optimal regions. A sensitivity analysis on surrogate accuracy could clarify whether this might be happening (specially with the more complex and dynamical tasks where the performances are mediocre)

- Empirical Results: While results in Figure 8 show slight improvements on simpler tasks (e.g., Hopper, Inverted Pendulum), Figure 15 shows that SAVO’s performance on more complex tasks (e.g., Ant, HalfCheetah) is only on par with other methods, without demonstrating a clear advantage.
Same trend is observed for results on recommender system in Figure 20 and Figure 22.

Overall the empirical results in this paper are not compelling enough to show a clear advantage of SAVO over existing methods. Across tasks, the results show that SAVO often performs on par with or worse than baseline methods, failing to convincingly outperform them. Therefore it’s unclear to me whether the proposed strategy is genuinely addressing suboptimal behavior or simply matching baseline performance without significant improvement. This leaves some doubt about the effectiveness of the approach in achieving better optimization in complex Q-landscapes.

**Questions:**

- How is the quality of the surrogate Q-landscape monitored to avoid filtering out high-reward regions? Is there a mechanism to re-evaluate excluded areas if necessary?
- Figure 9 Plot: The third plot in Figure 9 appears misaligned, with lines and text outside the figure’s bounds.

---

> ### Author Response · Authors · 2024-11-26
> **Added Sensitivity Analysis on Surrogate Loss**
>
> We sincerely thank the reviewer for their thoughtful evaluation and valuable feedback. We are pleased that you appreciated the identified problem, the novelty of our approach, and the convergence proof. We address your concerns below.
>
> ----
>
> ### 1. [Q] Quality of Action Proposals. [A] SAVO proposes better actions than baselines by using tabu-based surrogates
> > Quality of Action Proposals: I'm wondering whether the quality of action proposals from additional policies could affect the effectiveness of the maximizer actor? If these proposals are not close to the optimal actions, the maximizer actor’s effectiveness is limited, as it can only select from the actions provided.
>
> Precisely! Your comment aptly describes the capability and limits of the maximizer actor. Since one of the action proposals comes from the original actor, the maximizer actor is guaranteed to improve over it. However, the maximizer actor would only improve if the action proposals are of better quality. This is exactly why simple ways to propose actions such as *Ensemble* and *1-Actor k-Samples* baselines are not sufficiently effective in experiments (Figures 7a, 8).
>
> To address this limitation, SAVO introduces the idea of successive surrogates that reduce local optima (Section 4.2) and successive actors that propose effective actions by gradient-ascending these surrogates (Section 4.3). As a result, our proposed algorithm SAVO outperforms the baselines because **SAVO improves the effectiveness of the maximizer actor with higher quality action proposals.**
>
> While it is true that *achieving global optimum cannot be guaranteed*, SAVO significantly mitigates the suboptimality of the underlying DPG-actor with additional actor-surrogate pairs. Furthermore, SAVO’s quality of action proposals scales well with more actors and surrogates (Figure 11a,b). In conclusion, SAVO offers a gradient-based method that is fast at inference (unlike CEM), while finding near-optimal actions to improve over a single gradient-based actor. To our knowledge, **our work is the first to identify the challenge of local optima in gradient-based actors and propose an efficient solution to it**.
>
> ----
> ### 2. Added Surrogate Sensitivity Analysis: show surrogates stay up-to-date with the Q-function
> > The effect of smoothing in highly dynamical environments : In rapidly changing Q-landscapes, the surrogate-based tabu regions may become outdated, potentially causing the actor to miss newly optimal regions
>
> **[Surrogates are updated throughout training]**
>
> We clarify a potential misunderstanding about surrogates being outdated here. The surrogates are actually updated alongside the Q-function at every training step with Eq. 9.
>
> $$
> \mathcal{L}\_{\text{approx}} = \mathbb{E}\_{s \\sim \\rho^{\\mu}\_M} \left[ \sum\_{a \in \lbrace\tilde{\mu}\_M(s), \\nu\_i(s; a_{<i})\rbrace} \left\| \hat{\Psi}\_i(s, a; a\_{<i}) - \\Psi\_i(s, a; a\_{<i}) \right\|\_2^2 \right]
> $$
>
> So, whenever the Q-function is updated with a training sample $\(s_t, a_t, s_t’, r_t\)$, the surrogates are also updated at
> - (i) $a = a_t$, because that’s where the Q-function has latest updates $\implies$ **capture newly discovered optimal regions**, and
> - (ii) $a = \nu_i(s_k; a_{<i})$ because that’s where training the i-th actor $\nu_i$ requires a gradient $\implies$ make surrogate **up-to-date just before a gradient query** is made.
>
> &nbsp;
> &nbsp;
>
> > A sensitivity analysis on surrogate accuracy could clarify whether this might be happening (specially with the more complex and dynamical tasks where the performances are mediocre)
>
> **[Added sensitivity analysis of the surrogate accuracy: Figure 23]**
> Thank you for this great suggestion (also: Reviewer S3Pa)! We have added this in Appendix G.5 and plotted $\frac{\text{Surrogate Approximation Error}}{\text{Bellman Error}} \\%$ for all environments in Figure 23, where the numerator is $\mathcal{L}_\text{approx}$ from Eq. 9. We observe the following:
>
> - **Low Error:** Surrogates converge to 1—10\% of the Bellman error, successfully tracking Q-function changes.
> - **Non-zero error ensures smoothness:** Non-zero error ensures gradients propagate in flat regions, preventing stagnation, as proposed in Section 4.4.
>
> This analysis confirms surrogates remain accurate, simplify Q-landscapes, and enable robust gradient flow. Empirically, we showed that this approximation is necessary, because **SAVO beats its ablation without approximation underperforms in Figures 7b, 22**.
>
> &nbsp;
> &nbsp;
>
> > Q. How is the quality of the surrogate Q-landscape monitored to avoid filtering out high-reward regions? Is there a mechanism to re-evaluate excluded areas if necessary?
>
> We hope the above discussion clarifies that Equation (9) “re-evaluates” the surrogate at every step. No high-reward region from the Q-function is excluded because the information is propagated into the surrogate both (i) immediately and (ii) before a gradient query. This ensures that the surrogate remains accurate and up-to-date.

---

> > ### Author Response · Authors · 2024-11-26
> > **Added more empirical results on SAVO with SAC, and clarify current results**
> >
> > ### 3. [Q] SAVO Effectiveness? [A] SAVO successfully outperforms baselines whenever suboptimality exists due to non-convex Q-landscape
> > We thank the reviewer for referring to the exact figures to point out their source of concern. However, we clarify that *SAVO outperforms baselines in complex and simple tasks whenever the Q-landscape is severely non-convex*. If the baselines are not getting stuck in local optima, there is nothing for SAVO to improve, so it can only match their performance. However, we show that this problem is prevalent beyond simple benchmarks.
> >
> > **[Figure 7, 8 : Discrete Mine & RecSim, Restricted Locomotion]**
> > These are the tasks we identify as having non-convex Q-landscapes, and SAVO outperforms the underlying TD3 baseline significantly:
> > | Environment | TD3 | SAVO |
> > |---|---|---|
> > | MineWorld | 0.6 | **1.0** |
> > | RecSim | 4.7 | **5.5** |
> > | RecSim-Data | 43 | **50** |
> > | Restricted Hopper | 2100 | **2600** |
> > | Restricted Inv Pen | 700 | **950** |
> > | Restricted Inv Double Pen | **7200** | **7500** |
> > | Restricted Walker2D | 2800 | **3200** |
> >
> > **[Figure 10: Adroit Manipulation]**
> > While TD3 reaches optimal performance, SAVO requires fewer samples:
> > | Task   | SAVO Steps | Baseline Steps | Improvement |
> > |--------|------------|----------------|-------------|
> > | Door   | 1.5M       | 3M             | 50% faster  |
> > | Pen    | 2.5M       | 3M             | 16% faster  |
> > | Hammer | 2M         | 3M             | 33% faster  |
> >
> >
> > **[Figure 15 is “simpler” Unrestricted Locomotion]**
> > These are not “more complex tasks” but the standard unrestricted Mujoco family of tasks. The **baseline TD3 is already optimal** in these tasks because the Q-landscape is not severely non-convex.
> > - **Q-landscape visualization in unrestricted locomotion**: We visualize in Figures 29, 31 that Inverted Pendulum and Hopper Q-landscapes exhibit an easy path to the global optimum. Therefore, results show: $TD3 \approx TD3+SAVO$.
> > - **Q-landscape visualization in restricted locomotion**: However, when restrictions are added to these tasks, the Q-landscape becomes as shown in Figures 30, 32. These landscapes are significantly more non-convex, which results in TD3 being suboptimal in Figure 7, 8 results discussed above: $TD3 < TD3+SAVO$.
> >
> > Thus, SAVO outperforms baselines in Restricted locomotion but matches performance in unrestricted locomotion where there is no suboptimality to mitigate!
> >
> > **[Figure 20, 22 are Design Choice and Ablations of SAVO]**
> > We clarify that these are not “complex tasks”, but ablations of SAVO itself.
> > - Figure 20 shows that SAVO is robust to different design choices of list summarizers: Deep set, LSTM, Transformers — all perform well.
> > - Figure 22 shows that when each contribution of SAVO is removed, the performance slightly drops. This is to justify every technical contribution we make: (i) approximation in surrogate, (ii) condition on previous actions, (iii) removing TD3’s action smoothing (iv) successive architecture.
> >
> > **[Figure 19: SAVO improves in more complex RecSim tasks]**
> > When we increase the number of actions in RecSim to 100k and 500k, the suboptimality of baselines increases further, but SAVO remains robust, clearly outperforming the baselines.
> >
> > &nbsp;
> >
> >
> >
> > **Conclusion**: We recall that our contribution is to **mitigate the suboptimality of DPG-based methods** like T3. Thus, when there is no suboptimality, we do not expect SAVO to have any gains. However, we believe the results above convincingly demonstrate that SAVO outperforms baselines in complex tasks with non-convex Q-landscapes (Figures 7,8,10,19), while matching performance in simpler Q-landscapes (Figure 15).
> >
> > ----
> > ### 4. More empirical evidence for SAVO’s effectiveness
> > **[Added Figure 27: SAVO with more actors is significantly better]**
> >
> > While SAVO throughout uses only 3 actors, we show in Figure 27 that SAVO’s improvement over TD3 and other baselines is even more significant when the number of actors is optimized and chosen as 10 (or 15 in RecSim).
> >
> > **[Added Figure 28: SAC+SAVO > SAC in Ant, HalfCheetah]**
> >
> > Since TD3 already performs optimally in Ant and HalfCheetah, SAVO does not improve performance further. However, SAC suffers from a likewise local optima problem in these high-dimensional environments, and when combined with SAVO (described in Appendix J), the performance significantly improves.
> >
> > ----
> > > Figure 9 Plot: The third plot in Figure 9 appears misaligned, with lines and text outside the figure’s bounds.
> >
> > Thank you, we have now fixed Figure 9 in the rebuttal revision.
> >
> > ----
> >
> > We thank you for your valuable suggestions that helped refine our paper, and hope we've addressed your comments adequately. We are happy to answer any further questions and incorporate any further suggestions.

---

> ### Author Response · Authors · 2024-12-02
>
> We appreciate your time and effort to review our work. Since it is the last day of discussion, we would be grateful for any comments or suggestions in response to our rebuttal, and we would be happy to incorporate any specific suggestions. If our response adequately addresses your concerns, we kindly request to consider revising your score. Thank you.
>
> &nbsp;
>
> ----
>
> &nbsp;
>
>
> To summarize the key improvements made in response to your feedback:
> 1. We added **more empirical evidence**:
> - (i) SAVO improves further with more actors `(Fig. 27)`.
> - (ii) SAC + SAVO > SAC in Ant, HalfCheetah `(Fig. 28)`.
> 2. We added **surrogate approximation loss analysis** plot that shows SAVO stays up-to-date to true Q-value throughout training `(Fig. 23)`.
> 3. We emphasize **SAVO’s core contribution** is explicitly improving the quality of action proposals using tabu-based surrogates, thus beating naive maximizer-actor baselines `(Fig 7a, 8)`.
> 4. We highlight that **SAVO “mitigates” the suboptimality in non-convex Q-landscapes**, thus its gains are targeted to tasks where TD3 / SAC suffers. It outperforms TD3 significantly in non-convex Q-landscapes `(Fig 7,8,10,19)` and matches performance in simpler Q-landscapes `(Fig. 15)` because TD3 is already near-optimal there.

---

### Comment · Area_Chair_QrPo · 2024-11-23
**From AC**

Dear authors,
If possible, I wanted to ask you to start a discussion with the reviewers by providing a rebuttal.

Thanks,
AC

---

### Author Response · Authors · 2024-11-26
**Summary of revisions**

We sincerely thank the reviewers for feedback and suggestions for improvements. We apologize for the delayed rebuttal, as we conducted all the suggested analysis and extension experiments to ensure a comprehensive response. We hope the following revisions fully address your concerns and look forward to further discussion.

&nbsp;

**[Summarizing reviews]**
- The reviewers appreciated the importance of the central problem, novelty and scalability of the tabu-inspired SAVO approach, theoretical guarantees, paper writing, and extensive experiments on various benchmarks, baselines, and analyses.
- The primary feedback was to conduct **analyses** of (i) the approximation error of surrogates, (ii) the optimal number of actors needed, and (iii) SAVO’s performance on high-dimensional locomotion like Ant. Further suggestions included **extending** SAVO to SAC and enforcing diversity in SAVO.

&nbsp;

**[Summary of Revisions]**
We have exhaustively run the requested experiments, with updates to the paper highlighted under blue headings:
1. **Analyze surrogate approximation error** as a percentage of the Q-function bellman error across all tasks: `Appendix G.5`
$\implies$ the error stays small, meaning surrogates easily stay up-to-date to the Q-function, while being smooth.
2. **SAC+SAVO experiment**, which outperforms SAC in tasks like Ant and HalfCheetah: `Appendix J.`
$\implies$ TD3+SAVO $\approx$ TD3 > **SAC + SAVO > SAC**
3. **Find optimal number of actors** for SAVO to attain maximum performance in diverse environments: `Appendix I`
$\implies$ 10 actors are usually enough and SAVO more significantly outperforms baselines.
4. **Analyze Q-function smoothing**, motivated by smoothing of surrogates: `Appendix G.6`
$\implies$ Q-smoothing is essential for discrete-action tasks, and all our baselines already included it.
5. **Try diversity-enhancing initialization** of different actors and surrogates in SAVO: `Appendix G.7`
$\implies$ A promising idea, but default Xavier initialization works the best.

&nbsp;

**[Recap our contributions]**
We believe that our experiments sufficiently justify the primary contributions of our paper:
1. identify the problem of Deterministic Policy Gradient methods like TD3 are susceptible to the *actor getting stuck in local optima*,
2. study environments like restricted locomotion where TD3 actor gets stuck in suboptimal actions due to severe non-convexity in Q-function (unlike unrestricted Mujoco where TD3 is optimal),
3. visualize the non-convexity in Q-function landscapes,
4. propose a novel solution (SAVO) composed of a maximizer actor and tabu-inspired improvement that scales well with additional compute.
5. demonstrate local optima is mitigated via quantitative and visualized evidence on a variety of discrete and continuous action space tasks.
6. extensively validate ablations of SAVO and show it is robust to design choices.

---

### Author Response · Authors · 2024-12-04
**Summary of Rebuttal and Discussion**

We are grateful for the reviewers’ and AC’s time, and summarize our key discussions below.

&nbsp;

## Paper summary
Our paper identifies a key limitation in tasks with severely non-convex Q-functions, where an actor trained with deterministic policy gradients (e.g., TD3) is prone to becoming stuck in locally optimal actions, leading to suboptimal policy performance. We introduce SAVO as a novel actor architecture that mitigates this suboptimality through a sequence of additional actors and Q-like surrogates, enhancing performance without the need for extra hyperparameters.

&nbsp;

## Rebuttal summary
We received valuable suggestions on further analysis and extension experiments, which we incorporated as below:
1. `Surrogate Accuracy Analysis` [Appendix G.5]: The loss stays low showing surrogates follow the latest Q-function. (Reviewer PEvp, S3Pa)
2. `SAC Extension` [Appendix J]: SAC suffers suboptimality too; SAVO+SAC outperforms SAC in Ant, HalfCheetah. (Reviewer PEvp, S3Pa)
3. `Scalability Analysis` [Appendix I]: SAVO performance increases near-monotonically with more actor-surrogates until saturation,  thus a practical guideline is to use as many actors as feasible. However, even 2 additional actors help significantly. (Reviewer S3Pa, 6956)
4. `Smoothing and Diversity Extension` [Appendix G.6, G.7]: Smoothing Q-function helps in discrete-action tasks, wherein our baselines already perform Q-smoothing. We also tried the suggested diversity-enhanced initialization of the actors in SAVO, but did not see improvement. (Reviewer 6956)

&nbsp;

## Discussion summary
1. **`Reviewer 6956`** appreciated the experiments we added to their suggestion, especially on Q-smoothing and performance scaling with more actors, and subsequently raised their score.
2. **`Reviewer S3Pa`** acknowledged our comprehensive response with their suggested experiments on SAC+SAVO > SAC, surrogate accuracy, SAVO ablations, and design choices. Their one remaining concern was about the new analysis in Appendix I: *does SAVO introduce complexity for tuning the number of actors?*. We clarified that since SAVO performance gain is **near-monotonic with the number of actors**, there is no overhead needed to tune it — even our paper does not tune it and reports all results with **just two additional actors** — more would help further. Crucially, SAVO is guaranteed to improve the reliability of baselines like TD3 by trading off compute (quantified in Table 1) to provide substantial gains in performance and sample efficiency.

&nbsp;
We did not receive further engagement from the reviewer, but to address this potential confusion, we will revise the analysis title to be **“minimal” number of actors** for saturating SAVO performance, as the phrase “optimal number” might give the wrong indication that it is a sensitive hyperparameter that needs tuning.

3. **`Reviewer PEvp`** did not engage during rebuttal, but we incorporated their valuable suggestions on more empirical evidence to support SAVO. Their primary concern was about *why SAVO only matches the performance of TD3 in simpler unrestricted locomotion tasks in Figure 15?*. But, we clarified that this is exactly what is expected. SAVO is a **“suboptimality-mitigating” method** that matches TD3’s performance when the Q-landscape is not severely non-convex (visualizations in Figures 29, 31) because TD3 is already optimal, leaving no suboptimality to mitigate. However, when TD3/SAC is suboptimal due to severely non-convex Q-landscape (visualizations in Figures 30, 32), SAVO significantly outperforms the baselines as shown in Figures 7,8,10,19,28.

---

### Meta-Review · Area_Chair_QrPo · 2024-12-19

**Metareview:**

The paper attempts to improve policy gradient methods that rely on learning a critic. The authors (rightly) observe that gradient descent on the critic will not find the global optimum if the critic is non-convex. Instead, they propose optimising a series of surrogates.

The main strength of the paper is that it attempts to solve a hugely relevant problem (performance of methods that extend the deterministic policy gradient).

However, the paper has fatal weaknesses which make me doubt the quality of the proposed solution.
- Unsound theory. For example, near line 300 the universal approximation theorem is applied to a function that isn't continuous. Also, the equation near line 282 talks about gradients for $Q=\tau$, i.e. where the function isn't differentiable in general.
- Near line 305, evaluating at only two actions might not be enough data to fit a good surrogate.

For these reasons, I strongly recommend rejection.

If the authors want to resubmit, I would advise them to do the following.
- Remove all unsound theoretical claims
- Optimising a general non-convex function is (under mild assumptions) a very hard problem. The sampling inherent in running an RL algorithm only makes it harder. This means you should probably first address the case where the function you optimise is fully known. This would disentangle exploration and non-convex optimisation. Only when you can show an improvement in this regime would I advise applying it in the RL setting (and then I would recommend starting with bandits).

**Additional Comments On Reviewer Discussion:**

I strongly oppose accepting this paper (see meta-review for reasons).

Also, the review with the highest score is the lowest-quality.

---

### Decision · Program_Chairs · 2025-01-22

Reject